# Improved and Oracle-Efficient Online $\ell_1$-Multicalibration

**Rohan Ghuge** [1]  **Vidya Muthukumar** [2]  **Sahil Singla** [3]

## Abstract

We study *online multicalibration*, a framework for ensuring calibrated predictions across multiple groups in adversarial settings, across $T$ rounds. Although online calibration is typically studied in the $\ell_1$ norm, prior approaches to online multicalibration have taken the indirect approach of obtaining rates in other norms (such as $\ell_2$ and $\ell_\infty$) and then transferred these guarantees to $\ell_1$ at additional loss. In contrast, we propose a direct method that achieves improved and oracle-efficient rates of $\widetilde{\mathcal{O}}(T^{-1/3})$ and $\widetilde{\mathcal{O}}(T^{-1/4})$ respectively, for online $\ell_1$-multicalibration. Our key insight is a novel reduction of online $\ell_1$-multicalibration to an online learning problem with product-based rewards, which we refer to as *online linear-product optimization* (OLPO).

To obtain the improved rate of $\widetilde{\mathcal{O}}(T^{-1/3})$, we introduce a linearization of OLPO and design a no-regret algorithm for this linearized problem. Although this method guarantees the desired sublinear rate (nearly matching the best rate for online calibration), it is computationally expensive when the group family $\mathcal{H}$ is large or infinite, since it enumerates all possible groups. To address scalability, we propose a second approach to OLPO that makes only a polynomial number of calls to an offline optimization (*multicalibration evaluation*) oracle, resulting in *oracle-efficient* online $\ell_1$-multicalibration with a rate of $\widetilde{\mathcal{O}}(T^{-1/4})$. Our framework also extends to certain infinite families of groups (e.g., all linear functions on the context space) by exploiting a 1-Lipschitz property of the $\ell_1$-multicalibration error with respect to $\mathcal{H}$.

[1]H. Milton Stewart School of Industrial and Systems Engineering / Algorithms and Randomness Center, Georgia Institute of Technology, Atlanta, USA. [2]School of Electrical and Computer Engineering/H. Milton Stewart School of Industrial and Systems Engineering, Georgia Institute of Technology, Atlanta, USA. [3]School of Computer Science, Georgia Institute of Technology, Atlanta, GA, USA.. Correspondence to: Rohan Ghuge <rghuge3@gatech.edu>.

*Proceedings of the $42^{nd}$ International Conference on Machine Learning*, Vancouver, Canada. PMLR 267, 2025. Copyright 2025 by the author(s).

## 1. Introduction

Machine learning algorithms, powered by advances in data availability and model development, play a crucial role in decision-making across domains such as healthcare diagnostics, recidivism risk assessment, and loan approvals. This work focuses on forecasting algorithms that predict, say, the probability of binary outcomes $y$ (such as patient's severity or loan repayment) based on observable features $x$, in online settings where predictions are made as data is collected. A key metric used to evaluate the performance of such probability forecasters is *calibration* (Dawid, 1982). Roughly, it says that for any candidate prediction $p \in [0, 1]$, the fraction of forecasts with prediction $p$ should converge to $p$. In 1998, the seminal work of Foster & Vohra (1998) showed a bound of $O(T^{-1/3})$ for online calibration in the $\ell_1$ metric. Since then, this $O(T^{-1/3})$ bound has been reproved through insightful alternative approaches (Abernethy et al., 2011; Hart, 2022). However, improving this bound is a challenging open problem that has only recently seen some progress (Dagan et al., 2025; Qiao & Valiant, 2021).

Despite its popularity, calibration has a major limitation: calibrated predictions may perform poorly on specific subpopulations in the data, identifiable through contextual features such as gender, race, and age. To address this issue, Hébert-Johnson et al. (2018) proposed *multicalibration*, a framework designed to address discrimination arising from data in the batch setting. Informally, multicalibration is a requirement that the forecasts be statistically unbiased conditional both on its own prediction *and* on membership in any one of a large collection of intersecting subsets of the data space $\mathcal{H}$. Multicalibration and its variants have been an active area of research (see, e.g., Globus-Harris et al. (2023); Gupta et al. (2022); Jung et al. (2021); Kim et al. (2019; 2022)). Multicalibration has also found applications in *omniprediction* (Gopalan et al., 2022; 2023), a concept that asks for a single prediction which can be simultaneously used to optimize a large number of loss functions such that it is competitive with some hypothesis class $\mathcal{H}$. Approximately multicalibrated models in the $\ell_1$ metric turn out to automatically be omnipredictors, in both the batch (Gopalan et al., 2022) and online (Garg et al., 2024) settings.

We are especially focused on online multicalibration (Garg et al., 2024; Gupta et al., 2022), which naturally generalizes

the fundamental problem of sequential calibration. Existing approaches first provide multicalibration guarantees in the (weaker) $\ell_\infty$ or $\ell_2$ metric, and then transfer these guarantees to the $\ell_1$ metric at additional and possibly superfluous loss. The resulting rates for online $\ell_1$-multicalibration are significantly weaker than those for online calibration. A second drawback is that the runtime of most existing algorithms (with the exception of Garg et al. (2024), which we discuss later) for online multicalibration is typically linear in $|\mathcal{H}|$ (Gupta et al., 2022; Lee et al., 2022) . These algorithms are hence inefficient in even simple practical scenarios (e.g. linear functions in $d$ dimensions) where the hypothesis class is usually exponential in relevant problem parameters.

In light of these considerations, the main motivating question of our work is the following:

*Is there an online $\ell_1$-multicalibration algorithm that guarantees $O(T^{-1/3})$ error? Can we design "oracle-efficient" algorithms for online $\ell_1$-multicalibration?*

In this work, we make progress towards answering both these questions. We propose a method that achieves improved and oracle-efficient rates of $\widetilde{\mathcal{O}}(T^{-1/3})$ and $\widetilde{\mathcal{O}}(T^{-1/4})$, respectively, for online $\ell_1$-multicalibration. Our key insight is a novel reduction of online $\ell_1$-multicalibration to an online learning problem with product-based rewards, which we refer to as *online linear-product optimization* (OLPO). To obtain the improved rate of $\widetilde{\mathcal{O}}(T^{-1/3})$, we introduce a linearization of OLPO and design a no-regret algorithm for this linearized problem. Although this method guarantees the desired $\widetilde{\mathcal{O}}(T^{-1/3})$ rate, it becomes computationally expensive when the group family $\mathcal{H}$ is large or infinite, since it enumerates all possible groups. To address scalability, we propose a second approach to OLPO that makes only a single call per round to an offline optimization (*multicalibration evaluation*) oracle, resulting in *oracle-efficient* online $\ell_1$-multicalibration with a rate of $\widetilde{\mathcal{O}}(T^{-1/4})$. Our framework also extends to certain infinite families of groups (e.g., all linear functions on the context space) by exploiting a 1-Lipschitz property of the $\ell_1$-multicalibration error with respect to $\mathcal{H}$. We discuss the basic setup of online multicalibration, our results and our techniques in the rest of this section. Due to space limitations, we defer our treatment of related work to Appendix A.

## 1.1. Online Multicalibration

We now formally define the problem of online $\ell_1$-multicalibration (Garg et al., 2024; Gupta et al., 2022). Let $\mathcal{X}$ denote the context space and $\mathcal{Y} = [0, 1]$ denote the label domain, which we assume to be one-dimensional. Let $\mathcal{H}$ denote a collection of real-valued functions $h : \mathcal{X} \to \mathbb{R}$. We use $\mathcal{H}_B = \{h : \max_{\mathbf{x} \in \mathcal{X}} |h(\mathbf{x})| \le B\}$ to denote the set of functions with maximum absolute value on the context space bounded by $B$, and make the mild assumption that $\mathcal{H} \subseteq \mathcal{H}_B$

throughout the paper. For $n \in \mathbb{N}$, we use [n] to denote the set of integers $\{1, \ldots, n\}$. All of our algorithms will consider a discretized set of forecasts $\mathcal{P} := \{0, \frac{1}{m}, \ldots, 1\}$ for some discretization parameter $m \in \mathbb{N}$. We use $M$ to denote the size of $\mathcal{P}$, i.e., $M = |\mathcal{P}| = m + 1$.

Online prediction proceeds in rounds indexed by $t \in [T]$, for a given horizon of length $T$. In each round, the interaction between a learner and an adversary proceeds as follows:

1. The adversary selects a context $\mathbf{x}_t \in \mathcal{X}$ and a corresponding label $y_t \in \mathcal{Y}$.
2. The learner receives $\mathbf{x}_t$, but no information about $y_t$ is revealed.
3. The learner selects a distribution $\mathbf{w}_t$ and outputs $p_t \in \mathcal{P}$ sampled according to $\mathbf{w}_t$.
4. The learner observes $y_t$.

The learner's interaction with the adversary results in a *history* $\pi_T = \{\mathbf{x}_t, y_t, p_t\}_{t=1}^T$. We make no assumptions about the adversary; however, the learner is allowed to use randomness in making predictions. This induces a probability distribution over transcripts, and our goal is to design online algorithms that have low online $\ell_1$-multicalibration error in expectation, which is defined as follows:

**Definition 1.1** ($\ell_1$-Multicalibration Error). Given a transcript $\pi_T$, a function $h \in \mathcal{H}$, we define the learner's online $\ell_1$-multicalibration error with respect to $h$ as

$$K(\pi_T, h) := \sum_{p \in \mathcal{P}} \frac{1}{T} \left| \sum_{t \in S(\pi_T, p)} h(\mathbf{x}_t) \cdot (y_t - p_t) \right|,$$

where we define $S(\pi_T, p) = \{t \in [T] : p_t = p\}$. Finally, we define $\ell_1$-multicalibration error with respect to the family $\mathcal{H}$ as $K(\pi_T, \mathcal{H}) := \max_{h \in \mathcal{H}} K(\pi_T, h)$.

From here on, online multicalibration error will, by default, refer to online $\ell_1$-multicalibration error. When clear from context, we will drop $\pi_T$ and use $K(h)$ or $K(\mathcal{H})$ to denote the learner's online multicalibration error.

## 1.2. Our Results

Our first main result establishes an $\widetilde{\mathcal{O}}(T^{-1/3})$ rate for online $\ell_1$-multicalibration when the hypothesis class $\mathcal{H}$ is finite.

**Theorem 1.2.** *There is an algorithm that achieves online $\ell_1$-multicalibration error with respect to $\mathcal{H}$ with*

$$\mathbb{E}[K(\pi_T, \mathcal{H})] \le O\left(BT^{-1/3}\sqrt{\log(6T|\mathcal{H}|)}\right).$$

*The running time of this algorithm is linear in $|\mathcal{H}|$ and polynomial in $T$.*

This improves over an $\widetilde{\mathcal{O}}(T^{-1/4})$ bound obtained in Gupta et al. (2022) via online $\ell_\infty$-multicalibration (without needing to go through "bucketed" predictions), and nearly matches the best known bound for online calibration (Dagan et al., 2025).

**Comparison to Noarov et al. (2025).** After the initial submission of this work, it has been brought to our attention that a similar bound for online $\ell_1$-multicalibration for finite hypothesis class $\mathcal{H}$ could be derived from Theorem 3.4 of Noarov et al. (2025). We note that their algorithmic framework is quite different from ours, despite both works using an expert routine. In particular, their framework requires a small-loss regret bound to get the result, while a worst-case regret bound suffices in ours. Additionally, our algorithm is simpler (e.g. not requiring the solution of any min-max optimization problem) and proceeds through a novel reduction to the OLPO problem (defined in Section 2.1). We believe that this reduction is of independent interest as it facilitates the oracle-efficient results in a more natural and modular way.

Although Theorem 1.2 obtains an $\widetilde{\mathcal{O}}(T^{-1/3})$ rate for online $\ell_1$-multicalibration, it does not apply to infinite-sized hypothesis class $\mathcal{H}$. To address this, our next result obtains bounds in terms of the "covering number" of $\mathcal{H}$.

**Definition 1.3** ($\beta$-cover in $L_\infty$ metric, Bronshtein (1976)). For any function class $\mathcal{H}$, a finite subset of functions $\mathcal{H}_\beta = \{h^1, \ldots, h^N\} \subseteq \mathcal{H}$ is a $\beta$-cover with respect to the $L_\infty$ metric if for every $h \in \mathcal{H}$, there exists some $i \in [N]$ such that $\max_{\mathbf{x} \in \mathcal{X}} |h(x) - h^i(x)| \leq \beta$.

Let $\mathcal{H}_\beta$ denote a smallest possible $\beta$-cover for $\mathcal{H}$. We show that the online multicalibration error for infinite-sized hypothesis class $\mathcal{H}$ can be bounded in terms of $|\mathcal{H}_\beta|$.

**Theorem 1.4.** *There is an algorithm that achieves online $\ell_1$-multicalibration error with respect to $\mathcal{H}$ with*

$$\mathbb{E}[K(\pi_T, \mathcal{H})] \;\leq\; O\left(BT^{-1/3}\sqrt{\log(6T|\mathcal{H}_\beta|)}\right) + \beta.$$

*The running time of this algorithm is linear in $|H_\beta|$ and polynomial in $T$.*

As a consequence of this result, we obtain immediate applications to polynomial regression and bounded, Lipschitz convex functions (see Section 3.2 for details). We state one corollary here for the class of bounded linear functions.

**Corollary 1.5.** *Suppose $\mathcal{X} = [0, 1]^d$ and $\mathcal{H} = \{h \in \mathbb{R}^d : \|h\|_1 \leq B\}$ is the class of bounded linear functions. Then, there is an algorithm with runtime $O((B\sqrt{T})^d)$ that achieves online $\ell_1$-multicalibration error with respect to $\mathcal{H}$ with*

$$\mathbb{E}[K(\pi_T, \mathcal{H})] = \mathcal{O}\left(Bd^{1/2}T^{-1/3}\log(BT)\right).$$

Next, we give an "oracle-efficient" algorithm for online $\ell_1$-multicalibration for large hypothesis class $\mathcal{H}$. Our earlier algorithms need to enumerate over $\mathcal{H}$ or its $\beta$-cover, both of which are often exponentially large. We show how this can be avoided using the following *offline* oracle.

**Definition 1.6** (Offline Oracle). We receive a sequence of contexts $\{\mathbf{x}_s\}_{s=1}^t$ with corresponding reward vectors $\{\boldsymbol{f}_s\}_{s=1}^t$, coefficients $\{\kappa_s\}_{s=1}^t$, and an error parameter $\epsilon > 0$. The offline oracle returns a solution $(h^*, \boldsymbol{\theta}^*) \in \mathcal{H} \times \Theta$ that approximately solves, up to an additive error $\epsilon$,

$$\max_{h \in \mathcal{H}, \boldsymbol{\theta} \in \{\pm 1\}^M} \left\{ \textstyle\sum_{s=1}^t \kappa_s \langle \boldsymbol{\theta}, h(\mathbf{x}_s) \cdot \boldsymbol{f}_s \rangle \right\}.$$

Given such an oracle, our main result is the following.

**Theorem 1.7.** *There is an algorithm that achieves oracle-efficient online $\ell_1$-multicalibration with respect to a binary-valued $\mathcal{H} : \mathcal{X} \to \{0, 1\}$ with*

$$\mathbb{E}[K(\pi_T, \mathcal{H})] \leq \widetilde{O}\left(T^{-1/4}\sqrt{\log(T)}\right),$$

*under the assumptions of either transductive or sufficiently separated contexts (Syrgkanis et al., 2016) (see Section 4.2 for formal definitions). Moreover, this algorithm requires only a single call to the offline oracle per round.*

This result improves over the $\widetilde{O}(T^{-1/8})$ bound obtained in Garg et al. (2024), who also provide an oracle-efficient multicalibration algorithm but require access to an *online regression oracle*. Our oracle is equivalent to evaluating the online $\ell_1$-multicalibration error for a sequence of prediction in a one-shot manner — essentially, an *offline oracle*. Offline oracles are considered to be easier than online oracles. The assumptions in Theorem 1.7 on contexts and of binary-valued $\mathcal{H}$ are required to make the oracle *implementable* while maintaining the "stability" of the online algorithm, and are also commonly used in oracle-efficient online learning (Dudík et al., 2020; Syrgkanis et al., 2016). We also provide a generic "black-box" bound in Theorem 4.4.

### 1.3. Our Techniques

At the heart of our approach is a novel reduction of online $\ell_1$-multicalibration to an online learning problem with product-based rewards, which we refer to as *online linear-product optimization* (OLPO). In particular, we show that any learning algorithm for OLPO with regret $R_T(\mathcal{L}; \mathcal{H})$ can be efficiently transformed into an algorithm with online $\ell_1$-multicalibration error $R_T(\mathcal{L}; \mathcal{H})$, up to a small error (see Theorem 2.1). This reduction is crucial in both our improved and oracle-efficient rates for online $\ell_1$-multicalibration.

**Improved Rates.** Our first set of results establish the best-known information-theoretic rates for online $\ell_1$-multicalibration, beginning with the case of a finite-sized hypothesis class. We achieve this by designing a no-regret algorithm for OLPO. The main challenge in solving OLPO is that it is unclear *apriori* how to perform online optimization on a reward function that involves a product of variables. To address this, we define an online linear optimization problem in a higher-dimensional space and show that OLPO reduces to this problem, thereby effectively *linearizing* the

reward structure—at the cost of enumerating over all $h \in \mathcal{H}$. We denote this new problem as Lin-OLPO (or *linearized online linear-product optimization problem*). The reduction from OLPO to Lin-OLPO introduces two issues. The first issue is that we need to ensure that the best fixed actions in OLPO and Lin-OLPO remain consistent. To address this, we introduce a specific mixed norm and restrict the set of actions of Lin-OLPO to a unit ball in this mixed norm. The second issue is that the expanded decision space may introduce better actions for Lin-OLPO that are not valid in OLPO. To resolve this, we appropriately scale decisions as we translate them from Lin-OLPO to OLPO, ensuring consistency between the action spaces of OLPO and Lin-OLPO (see Lemma 3.2).

Next, we design a no-regret algorithm for Lin-OLPO with the mixed-norm constraint that carefully combines two components: (i) a reward-maximizing no-regret online linear optimization (OLO) algorithm (e.g., online gradient descent), and (ii) a reward-maximizing no-regret algorithm for the experts setting (e.g., multiplicative weights update). Our approach runs $|\mathcal{H}|$ instances of the OLO algorithm in parallel. Each OLO instance executes the action optimal for calibration with respect to a specific hypothesis $h \in \mathcal{H}$. Meanwhile, the experts algorithm identifies hypotheses that appear "more difficult" with respect to calibration error—where difficulty is measured using the cumulative reward of the corresponding OLO algorithm. We obtain the final guarantee by analyzing the regret of this algorithm (see Lemma 3.3) and combining it with the two earlier reductions.

To transfer these bounds to an infinite-sized hypothesis class $\mathcal{H}$ (Theorem 1.4), we first show that online multicalibration error is 1-Lipschitz with respect to $\mathcal{H}$. Then, we construct an appropriate *covering* of the hypothesis class $\mathcal{H}$, and appeal to online $\ell_1$-multicalibration rates for finite $\mathcal{H}$.

**Oracle Efficiency.** A plethora of oracle-efficient online learning algorithms have been developed in the last decade, with the aim of efficient online regret minimization given an *offline* optimization oracle (Daskalakis & Syrgkanis, 2016; Dudík et al., 2020; Syrgkanis et al., 2016). We cannot easily adapt these frameworks to Lin-OLPO, primarily because the algorithm needs to maintain and access $|\mathcal{H}|$ parallel copies of OLO algorithms, even if Step (ii) above is made efficient. Fundamentally, Lin-OLPO operates in an augmented space that is linear in $|\mathcal{H}|$ and therefore intractable.

We instead adapt the oracle-efficient framework directly to the OLPO problem, circumventing the need to access the augmented Lin-OLPO structure. We leverage the *generalized Follow-the-Perturbed-Leader* family of algorithms (Dudík et al., 2020) and show that, remarkably, a regret bound for OLPO can be derived using similar techniques as in Dudík et al. (2020) using certain special properties of the OLPO

structure. Specifically, it suffices to restrict decisions for OLPO to the Boolean hypercube; i.e., $\{\pm 1\}^M$. This allows us to ultimately prove Theorem 1.7.

## 2. Reducing Multicalibration to OLPO

In this section, we show how to efficiently reduce online multicalibration to an online learning problem with product-based rewards, which we refer to as *online linear-product optimization* (OLPO), and define next. This reduction will be crucial to our improved rates for online multicalibration. We note that there is precedent for connections between calibration and regret; in particular, (Abernethy et al., 2011) provided a simpler reduction between calibration and online linear optimization.

### 2.1. Online Linear-Product Optimization

We formally define the *online linear-product optimization problem* (OLPO). Let $\mathcal{X}$ denote the context space and let hypothesis class $\mathcal{H} \subseteq \mathcal{H}_B$ be a collection of $B$-bounded real-valued functions $h : \mathcal{X} \to \mathbb{R}$. Let $\mathbb{B}_\infty \subseteq \mathbb{R}^M$ denote the unit cube, i.e., $\mathbb{B}_\infty = \{\boldsymbol{\theta} \in \mathbb{R}^M : \|\boldsymbol{\theta}\|_\infty \leq 1\}$. The set $\mathcal{H} \times \mathbb{B}_\infty$ will denote an action set. In each round $t \in [T]$:

1. Learner plays a function $h_t \in \mathcal{H}$ and vector $\boldsymbol{\theta}_t \in \mathbb{B}_\infty$.
2. Adversary then reveals a context $\mathbf{x}_t$ and a reward vector $\boldsymbol{f}_t \in \mathbb{R}^M$.
3. Learner then receives reward $\langle \boldsymbol{\theta}_t, h_t(\mathbf{x}) \cdot \boldsymbol{f}_t \rangle$.

Note that this is not a standard online linear optimization problem since it involves a product of variables. We will use $\mathcal{L}$ to denote a generic algorithm for OLPO. This algorithm takes as input a sequence of vectors $(\mathbf{x}_1, \boldsymbol{f}_1), \ldots, (\mathbf{x}_{t-1}, \boldsymbol{f}_{t-1})$ and returns a pair $(h_t, \boldsymbol{\theta}_t) \in \mathcal{H} \times \mathbb{B}_\infty$. We denote by $R_T(\mathcal{L}; (\mathbf{x}_1, \boldsymbol{f}_1) \ldots, (\mathbf{x}_T, \boldsymbol{f}_T); \mathcal{H})$ the regret of $\mathcal{L}$ when compared to the best fixed action $(h^*, \boldsymbol{\theta}^*)$. Formally, $R_T(\mathcal{L}; (\mathbf{x}_1, \boldsymbol{f}_1) \ldots, (\mathbf{x}_T, \boldsymbol{f}_T); \mathcal{H})$ equals: $\max_{h^* \in \mathcal{H}, \boldsymbol{\theta}^* \in \mathbb{B}_\infty} \left\{ \sum_{t=1}^{T} h^*(\mathbf{x}_t) \cdot \langle \boldsymbol{\theta}^*, \boldsymbol{f}_t \rangle \right\} - \sum_{t=1}^{T} h_t(\mathbf{x}_t) \cdot \langle \boldsymbol{\theta}_t, \boldsymbol{f}_t \rangle$. When the input sequence is clear, we will omit it from the definition and simply write $R_T(\mathcal{L}; \mathcal{H})$.

Recall that in online multicalibration, the learner receives a context $\mathbf{x}_t$ in each round $t \in [T]$ and makes a prediction $p_t \in \mathcal{P} = [1/m]$ according to some distribution $\mathbf{w}_t$. Then, $\mathbf{w}_t$ is a $M$-dimensional probability vector (recall that $M = |\mathcal{P}|$), where $\mathbf{w}_t(i)$ represents the probability that $p_t$ equals $i/m$. The main result of this section shows that an algorithm for OLPO can be efficiently converted into an algorithm for online $\ell_1$-multicalibration.

**Theorem 2.1.** *Let $\mathcal{L}$ be an algorithm for OLPO for some collection $\mathcal{H} \subseteq \mathcal{H}_B$ with expected regret denoted by $R_T(\mathcal{L}; \mathcal{H})$. Then, there is a sequential prediction algorithm with online*

$\ell_1$-multicalibration error with respect to $\mathcal{H}$ bounded by

$$\frac{B}{m} + \frac{R_T(\mathcal{L};\mathcal{H})}{T} + 4B\sqrt{\frac{m\log(6T|\mathcal{H}|)}{T}} + \frac{4mB}{T}. \qquad (1)$$

*Moreover, this algorithm is efficient; its running time is polynomial in the running time of $\mathcal{L}$, the discretization parameter $m$, and $T$.*

We will instantiate Theorem 2.1 with different online learning algorithms for OLPO in Section 3 and Section 4. In the remainder of this section, we outline the reduction and proof sketch of Theorem 2.1. In this reduction, the context space $\mathcal{X}$ and the collection of functions $\mathcal{H}$ remain unchanged. The key steps will involve utilizing the fact that the actions are in $\mathcal{H} \times \mathbb{B}_\infty$ and carefully setting the reward vectors $\boldsymbol{f}_1, \boldsymbol{f}_2, \ldots, \boldsymbol{f}_T \in \mathbb{R}^M$.

## 2.2. The Reduction

In order to convert an algorithm for OLPO into an online multicalibration algorithm, we need a "halfspace oracle". One such oracle was used to reduce calibration to Blackwell approachability (and subsequently to no-regret learning) in (Abernethy et al., 2011).

**Definition 2.2** (Halfspace Oracle). We assume access to an efficient halfspace oracle $\mathcal{O}$ that can, given $\mathbf{x} \in \mathcal{X}$, $h \in \mathcal{H}$ and $\boldsymbol{\theta} \in \mathbb{B}_\infty$, select a probability distribution $\mathbf{w} \in \mathbb{R}^M$ with $\|\mathbf{w}\|_1 = 1$ such that for all $y \in \mathcal{Y}$, we have

$$\textstyle\sum_{i=0}^{m} \boldsymbol{\theta}(i) \cdot h(\mathbf{x}) \cdot \mathbf{w}(i) \cdot \left(y - \frac{i}{m}\right) \ \leq \ \frac{B}{m}.$$

A surprising result of (Abernethy et al., 2011) shows that an efficient halfspace oracle always exists in the context of calibration for $\mathcal{Y} = [0, 1]$. We show that this result extends to our setting, i.e., given $h$ and $\mathbf{x}$, the oracle construction remains unchanged. See Appendix B.3 for details.

**Lemma 2.3** (Algorithm 4). *Given any $\mathbf{x} \in \mathcal{X}$, $h \in \mathcal{H}$, and $\boldsymbol{\theta} \in \mathbb{B}_\infty$, there exists an efficient halfspace oracle.*

We are now ready to describe our multicalibration algorithm.

**Multicalibration Algorithm.** At each round $t \in [T]$, the algorithm randomly predicts $p_t$ according to some distribution $\mathbf{w}_t \in \mathbb{R}^M$. The distribution $\mathbf{w}_t$ is obtained using a combination of the output of the learning algorithm $\mathcal{L}$ for OLPO and the halfspace oracle $\mathcal{O}$. In particular, given previous contexts $\mathbf{x}_1, \ldots, \mathbf{x}_{t-1}$ and previous vectors $\boldsymbol{f}_1, \ldots, \boldsymbol{f}_{t-1}$, let $(h_t, \boldsymbol{\theta}_t)$ denote the action selected by $\mathcal{L}$ in round $t$. The prediction distribution for current context $\mathbf{x}_t$ is now obtained using the halfspace oracle: $\mathbf{w}_t = \mathcal{O}(\mathbf{x}_t, h_t, \boldsymbol{\theta}_t)$. Finally, after observing $y_t$, define $\boldsymbol{f}_t := \boldsymbol{f}_t(\mathbf{w}_t, y_t)$ where

$$\boldsymbol{f}_t(\mathbf{w}_t, y_t)_i = \mathbf{w}_t(i) \cdot \left(y_t - \frac{i}{m}\right). \qquad (2)$$

We formally describe the reduction in Algorithm 1.

---

**Algorithm 1** ONLINE $\ell_1$-MULTICALIBRATION

1: **for** $t = 1, \ldots T$ **do**
2:     observe $\mathbf{x}_t$.
3:     query the OLPO algorithm:
    $(h_t, \boldsymbol{\theta}_t) \leftarrow \mathcal{L}((\mathbf{x}_1, \boldsymbol{f}_1), \ldots, (\mathbf{x}_{t-1}, \boldsymbol{f}_{t-1}))$.
4:     query the halfspace oracle: $\mathbf{w}_t \leftarrow \mathcal{O}(\mathbf{x}_t, h_t, \boldsymbol{\theta}_t)$.
5:     predict $p_t \sim \mathbf{w}_t$ and observe $y_t$.
6:     $\boldsymbol{f}_t \leftarrow \boldsymbol{f}_t(\mathbf{w}_t, y_t)$ as per (2)
7: **end for**

---

**Proof Sketch of Theorem 2.1.** We first relate the expected multicalibration error for any group $h$ to $\|\frac{1}{T}\sum_{t=1}^{T} h(\mathbf{x}_t) \cdot \boldsymbol{f}_t\|_1$ through a martingale argument (see (8)). Then, using the definition of the dual norm,

$$\left\|\frac{1}{T}\sum_{t=1}^{T} h(\mathbf{x}_t) \cdot \boldsymbol{f}_t\right\|_1 = \frac{1}{T}\sup_{\|\boldsymbol{\theta}\|_\infty \leq 1}\left\langle \boldsymbol{\theta}, \sum_{t=1}^{T} h(\mathbf{x}_t) \cdot \boldsymbol{f}_t\right\rangle.$$

Thus, the overall multicalibration error can be (roughly) related to the reward of the corresponding instance of OLPO. We include the complete proof of Theorem 2.1 in Appendix B.1.

## 3. Improved Online Multicalibration

In this section, we give upper bounds for online multicalibration foregoing computational complexity considerations. In Section 3.1, we consider the case of a finite hypothesis class $\mathcal{H}$ and provide upper bounds through the design of an appropriate no-regret algorithm for OLPO. Combined with Theorem 2.1 and Lemma 3.2, this yields sublinear bounds for online multicalibration when the hypothesis class $\mathcal{H}$ is finite, proving Theorem 1.2. In Section 3.2, we describe the changes necessary to handle an infinite-sized hypothesis class $\mathcal{H}$ and prove Theorem 1.4.

### 3.1. Online Multicalibration for Finite Groups

We describe a no-regret algorithm for OLPO in the case where the hypothesis class $\mathcal{H}$ is finite. The key idea is to introduce an online linear optimization problem in a higher-dimensional space, and to subsequently reinterpret OLPO in terms of this problem. We denote this problem as Lin-OLPO (or *linearized online linear-product optimization problem*). We set up some preliminaries to define Lin-OLPO. We index the elements of $\mathcal{H}$ by $h^{(1)}, h^{(2)}, \ldots h^{|\mathcal{H}|}$. Given a vector $\mathbf{z} \in \mathbb{R}^{|\mathcal{H}| \times M}$, let $\mathbf{z}(h)$ denote the $M$-dimensional sub-vector of $\mathbf{z}$ indexed by $h \in \mathcal{H}$. Then, we define the mixed norms:

$$\|\mathbf{z}\|_{1,\infty} := \sum_{h \in \mathcal{H}} \|\mathbf{z}(h)\|_\infty \quad \text{and} \quad \|\mathbf{z}\|_{\infty,1} := \max_{h \in \mathcal{H}} \|\mathbf{z}(h)\|_1$$

and use $\mathbb{B}_{1,\infty} = \{\widetilde{\boldsymbol{\theta}} : \|\widetilde{\boldsymbol{\theta}}\|_{1,\infty} \leq 1\}$ and $\mathbb{B}_{\infty,1} = \{\widetilde{\boldsymbol{f}} : \|\widetilde{\boldsymbol{f}}\|_{\infty,1} \leq 1\}$ to denote a unit in the respective norms.

We now define Lin-OLPO, which is an online linear optimization problem over $\mathbb{B}_{1,\infty}$.

**Definition 3.1** (Lin-OLPO). In round $t \in [T]$ of Lin-OLPO, the learner plays (a possibly random) action $\widetilde{\boldsymbol{\theta}}_t \in \mathbb{B}_{1,\infty}$. The adversary reveals a reward vector $\widetilde{\boldsymbol{f}}_t \in \mathbb{B}_{\infty,1}$, which leads to a *linear* reward $\left\langle \widetilde{\boldsymbol{\theta}}_t, \widetilde{\boldsymbol{f}}_t \right\rangle$. The goal of an online learning algorithm for Lin-OLPO, denoted henceforth by $\widetilde{\mathcal{L}}$, is to minimize the regret. The regret, denoted by $R_T^{\texttt{Lin-OLPO}}(\widetilde{\mathcal{L}}; B_{1,\infty})$, equals:

$$\max_{\widetilde{\boldsymbol{\theta}} \in B_{1,\infty}} \left\langle \widetilde{\boldsymbol{\theta}}, \sum_{t=1}^{T} \widetilde{\boldsymbol{f}}_t \right\rangle - \sum_{t=1}^{T} \mathbb{E}\left[ \left\langle \widetilde{\boldsymbol{\theta}}_t, \widetilde{\boldsymbol{f}}_t \right\rangle \right],$$

where the expectation is taken over possible randomness in $\widetilde{\boldsymbol{\theta}}_t$.

We now present a reduction from OLPO to Lin-OLPO. Recall that the primary challenge with OLPO is that the reward function involves a product of variables. However, by appropriately "expanding" the reward vectors $\boldsymbol{f}_1, \boldsymbol{f}_2, \dots$, and the decision space to cover all possible $h \in \mathcal{H}$, we can effectively *linearize* the reward function, aligning it with the structure of Lin-OLPO. This linearization introduces two key challenges: (1) the need to ensure that the best fixed actions in OLPO and Lin-OLPO remain consistent, and (2) preventing the expanded decision space in Lin-OLPO from introducing "better" actions that cannot be captured by OLPO (see Section 1.3 for a more detailed discussion).

**Lemma 3.2.** *Let $\widetilde{\mathcal{L}}$ be an online learning algorithm for* Lin-OLPO *with corresponding expected regret $R_T^{\texttt{Lin-OLPO}}(\widetilde{\mathcal{L}}; \mathbb{B}_{1,\infty})$. Then, there exists a randomized learning algorithm $\mathcal{L}$ for* OLPO *with the decision set $\mathcal{H}_B \times \mathbb{B}_\infty$ such that its expected regret $R_T(\mathcal{L}; \mathcal{H}_B) = B \cdot L \cdot R_T^{\texttt{Lin-OLPO}}(\widetilde{\mathcal{L}}; \widetilde{\mathbb{B}}_{1,\infty})$, where $L = \max_{t \in [T]}\{\|\boldsymbol{f}_t\|_1\}$ is the maximum $\ell_1$-norm of the reward vectors in the* OLPO *instance.*

We defer the proof of this lemma to Appendix C.1.

We now present a no-regret algorithm for Lin-OLPO whose runtime per round is linear in $|\mathcal{H}|$.

**Algorithm.** Our algorithm for Lin-OLPO relies on two components: (i) a reward-maximizing no-regret online linear optimization algorithm, denoted by $\mathcal{A}$ (e.g., online gradient descent), and (ii) a reward-maximizing no-regret algorithm for the experts setting, denoted $\mathcal{E}$ (e.g., multiplicative weights update). Our algorithm will execute $|\mathcal{H}|$ copies of $\mathcal{A}$, one for each $h \in \mathcal{H}$; that is, we treat each function $h$ as an expert who runs their own copy of $\mathcal{A}$, denoted $\mathcal{A}^h$, each measuring regret against the action set $\mathbb{B}_\infty$. Subsequently, the predictions made by $\mathcal{A}^h$ are aggregated using $\mathcal{E}$. At a high level, each of the OLO algorithms $\mathcal{A}^h$ executes the action we would want to take if we were only concerned about calibration with respect to the specific hypothesis $h$, and the experts algorithm $\mathcal{E}$ selects for hypotheses that seem "more difficult" with respect to calibration error (where our proxy

for calibration error is precisely the cumulative reward of the corresponding OLO algorithm). See Algorithm 2 for a formal description.

---

**Algorithm 2** LINEARIZED ONLINE LINEAR-PRODUCT OPTIMIZATION

1: **Input:** OLO algorithm $\mathcal{A}$, experts algorithm $\mathcal{E}$
2: start instances $\mathcal{A}^1, \dots, \mathcal{A}^{|\mathcal{H}|}$, one for each $h \in \mathcal{H}$
3: **for** $t = 1, \dots T$ **do**
4:     obtain prediction $\boldsymbol{\theta}_t^h$ from $\mathcal{A}^h$, for $h \in \mathcal{H}$
5:     query $\mathcal{E}$ to obtain distribution $\boldsymbol{\gamma}_t$
6:     let $h_t = h$ with probability $\boldsymbol{\gamma}_t(h)$; predict $\boldsymbol{\theta}_t = \boldsymbol{\theta}_t^{h_t}$
7:     observe reward vector $\widetilde{\boldsymbol{f}}_t$
8:     for $h \in \mathcal{H}$, pass reward $\widetilde{\boldsymbol{f}}_t(h)$ to $\mathcal{A}^h$
9:     for $h \in \mathcal{H}$, pass reward $\langle \boldsymbol{\theta}_t^h, \widetilde{\boldsymbol{f}}_t(h) \rangle$ to $\mathcal{E}$
10: **end for**

---

In Algorithm 2, we will use the popular *online gradient descent* algorithm as our reward-maximizing no-regret online linear optimization algorithm, which is known to have regret $O(DG/\sqrt{T})$ where $D$ denotes the $\ell_2$ diameter of the action set and $G$ denotes the $\ell_2$-norm of the largest reward vector (Hazan et al., 2016). For the experts algorithm $\mathcal{E}$, we use the familiar *multiplicative weights update* algorithm which provides regret $O(\rho\sqrt{Tn})$ when rewards are bounded in $[-\rho, \rho]$ and $n$ denotes the number of experts. Consequently, we obtain the following guarantee.

**Lemma 3.3.** *The regret $R_T^{\texttt{Lin-OLPO}}(\widetilde{\mathcal{L}}; \mathbb{B}_{1,\infty})$ of Algorithm 2 can be bounded as follows.*

$$R_T^{\texttt{Lin-OLPO}}(\widetilde{\mathcal{L}}; \mathbb{B}_{1,\infty}) \leq \sqrt{T \log |\mathcal{H}|} + \sqrt{TM}.$$

**Proof Sketch.** At a high level, the regret bound proceeds by first applying the regret bound for the multiplicative weights update algorithm, and then subsequently applying the no-regret property given by each instantiation, i.e., for each $h \in \mathcal{H}$, of the online gradient descent algorithm. We provide the detailed proof in Appendix C.2.

We can now complete the proof of Theorem 1.2.

*Proof of Theorem 1.2.* Using Theorem 2.1, Lemma 3.2 and Lemma 3.3, $\mathbb{E}[K(\pi_T, \mathcal{H})] \leq 7BT^{-1/3}\sqrt{\log(6T|\mathcal{H}|)} + 5BT^{-1/2}\sqrt{\log|\mathcal{H}|} = O\left(BT^{-1/3}\sqrt{\log(6T|\mathcal{H}|)}\right)$, where the penultimate inequality uses $M \leq 2m$, and the final inequality uses $L \leq 1$ as per the reduction in Theorem 2.1 and $m = T^{1/3}$. $\square$

### 3.2. Extending to Infinitely Many Groups
We now describe the necessary changes to obtain improved bounds for online multicalibration for infinite-sized hypothesis class $\mathcal{H}$. At a high-level, our approach reduces to the finite setting by constructing an appropriate *covering* of the

hypothesis class $\mathcal{H}$. Then, it uses the simple fact that the online multicalibration error is 1-Lipschitz w.r.t. $\mathcal{H}$.

Recall our definition of covering with respect to the $L_\infty$ metric (Definition 1.3). We say that a subset $\mathcal{H}_\beta = \{h^1, \ldots, h^N\} \subseteq \mathcal{H}$ is a $\beta$-cover with respect to the $L_\infty$ metric if for every $h \in \mathcal{H}$, there exists some $i \in [N]$ such that $\|h - h^i\|_{L_\infty} = \max_{x \in \mathcal{X}} |h(x) - h^i(x)| \leq \beta$. Henceforth, we denote $\mathcal{H}_\beta$ and $|\mathcal{H}_\beta|$ as the minimal $\beta$-cover and the corresponding $\beta$-covering number respectively of $\mathcal{H}$, referring to the $\ell_\infty$ metric by default. With this definition in hand, the key steps for our extension to an infinite-sized hypothesis class $\mathcal{H}$ are as follows:

1. We begin by replacing $\mathcal{H}$ with $\mathcal{H}_\beta$ for some $\beta > 0$ and subsequently appealing to Theorem 1.2 to bound the online multicalibration error with respect to $\mathcal{H}_\beta$.

2. Then, by showing that the online multicalibration error is 1-Lipschitz with respect to $\mathcal{H}$ in the $L_\infty$ metric, we can conclude that the prediction is indeed multicalibrated with respect to $\mathcal{H}$ up to the additional error term $\beta$ (see Lemma 3.4).

3. Finally, we appropriately set $\beta = \frac{o(T)}{T}$ (in a manner that optimally balances the terms $\sqrt{\log |\mathcal{H}_\beta|}$ and $\beta$ in the multicalibration error). We instantiate the last step with various examples of linear, polynomial, and uniformly Lipschitz convex function classes.

The main result of this section shows that *any* algorithm that is multicalibrated with respect to $\mathcal{H}_\beta$ is also multicalibrated with respect to $\mathcal{H} \subseteq \mathcal{H}_B$ up to an additional error of $\beta$.

**Lemma 3.4.** *Let $\mathcal{H}$ denote a collection of real-valued functions $h : \mathcal{X} \to \mathbb{R}$ and let $\mathcal{H}_\beta$ denote a $\beta$-cover of $\mathcal{H}$ with respect to the $L_\infty$ metric. If a sequence of predictions is multicalibrated with respect to $\mathcal{H}_\beta$, then it is also multicalibrated with respect to $\mathcal{H}$, up to an additive $\beta$. Formally, if $K(\pi_T, \mathcal{H}_\beta) \leq \alpha$, then $K(\pi_T, \mathcal{H}) \leq \alpha + \beta$.*

The proof of this lemma can be found in Appendix C.3. We can now complete the proof of Theorem 1.4 using Theorem 1.2 and Lemma 3.4.

*Proof of Theorem 1.4.* Fix an $h \in \mathcal{H}$, and let $h' \in \mathcal{H}_\beta$ such that $\max_{\mathbf{x} \in \mathcal{X}} |h(\mathbf{x}) - h'(\mathbf{x})| \leq \beta$. Then, $\mathbb{E}[K(\pi_T, h)] = \mathbb{E}[K(\pi_T, h')] + \mathbb{E}[K(\pi_T, h) - K(\pi_T, h')] \leq O(BT^{-1/3}\sqrt{\log(6T|\mathcal{H}_\beta|)}) + \beta$, where the first term is bounded by Theorem 1.2 on the class $\mathcal{H}_\beta$ and the second term follows from Lemma 3.4 since $\mathcal{H}_\beta$ is a $\beta$-cover. $\square$

We apply this result to specific function classes that are known to have bounded covering numbers, namely, to polynomial regression and to bounded, uniformly Lipschitz, convex functions. The class of bounded linear functions is subsumed by the polynomial regression class (when $k = 1$),

for which we obtain Corollary 1.5. See Appendix C.4 for details.

# 4. Oracle Efficient Online Multicalibration

In this section, we explore approaches to computationally efficient online multicalibration and ultimately prove Theorem 1.7. We do this by adopting the framework of *oracle efficiency* (Daskalakis & Syrgkanis, 2016; Hazan & Koren, 2016; Syrgkanis et al., 2016) for the online linear-product optimization problem (OLPO) arising from the reduction in Theorem 2.1. In particular, we design an online learning algorithm for OLPO that makes a single call to an optimization oracle per round[1] and does not need to access the augmented Lin-OLPO structure. We recall the definition of our oracle.

**Definition 4.1** (Offline Oracle). The offline oracle receives a sequence of contexts $\{\mathbf{x}_s\}_{s=1}^t$ with corresponding reward vectors $\{\boldsymbol{f}_s\}_{s=1}^t$, coefficients $\{\kappa_s\}_{s=1}^t$, and an error parameter $\epsilon > 0$. The oracle returns a solution $(h^*, \boldsymbol{\theta}^*) \in \mathcal{H} \times \Theta$ that approximately solves, up to an additive error $\epsilon$,

$$\max_{h \in \mathcal{H}, \boldsymbol{\theta} \in \Theta} \left\{ \sum_{s=1}^t \kappa_s \langle \boldsymbol{\theta}, h(\mathbf{x}_s) \cdot \boldsymbol{f}_s \rangle \right\}. \tag{3}$$

Note that the oracle can handle input of variable length, i.e., the number of rounds $t$ itself is an implicit input to the optimization oracle.

In the introduction, we claimed that solving (3) essentially amounts to evaluating approximate multicalibration error. To see this, suppose we call the oracle with input length equal to $T$ and set $\kappa_t = 1$ for all $t \leq T$. Then, for a fixed $h \in \mathcal{H}$, we have $\max_{\boldsymbol{\theta} \in \Theta} \sum_{t=1}^T \langle \boldsymbol{\theta}, h(\mathbf{x}_t) \cdot \boldsymbol{f}_t \rangle = \max_{\boldsymbol{\theta} \in \Theta} \langle \boldsymbol{\theta}, \sum_{t=1}^T h(\mathbf{x}_t) \cdot \boldsymbol{f}_t \rangle = \left\| \sum_{t=1}^T h(\mathbf{x}_t) \cdot \boldsymbol{f}_t \right\|_1$, and the argument $\boldsymbol{\theta}$ that maximizes the above is expressible in closed form. Maximizing this quantity over $h \in \mathcal{H}$ yields the approximation to the multicalibration error with respect to the family $\mathcal{H}$ that is defined in Equation (8) (where the approximation error is upper bounded in Lemma B.1).

In the remainder of this section, we design an oracle-efficient online learning algorithm for OLPO that relies on the oracle defined in (3). In Section 4.2, we provide explicit efficient constructions of the oracle under the assumption that the contexts $\{\mathbf{x}_t\}_{t=1}^T$ satisfy either the *transductive* or *small-separator* conditions, both of which are commonly made assumptions in oracle-efficient online learning (Dudík et al., 2020; Syrgkanis et al., 2016). We restate these assumptions for completeness in Section 4.2.

We first make an observation tailored to the decision set $B_\infty$: recall that in round $t \in [T]$, the learner in OLPO plays

---

[1]Note that we can think of oracle calls requiring $O(1)$ time, as we have dispatched computational effort to the optimization oracle.

$(h_t, \boldsymbol{\theta}_t) \in \mathcal{H} \times B_\infty$. For any $h \in \mathcal{H}$, we have

$$\arg\max_{\boldsymbol{\theta} \in B_\infty} \left\{ \sum_{t=1}^{T} h(\mathbf{x}_t) \cdot \langle \boldsymbol{f}_t, \boldsymbol{\theta} \rangle \right\} \in \{\pm 1\}^M.$$

Based on this observation, we restrict our decisions to the Boolean hypercube; i.e., $\boldsymbol{\theta}_t \in \{\pm 1\}^M$, and note that for any learning algorithm $\mathcal{L}$ for OLPO, the regret remains unchanged. We also allow actions $\boldsymbol{\theta}_t$ to be randomized; as a result, $\mathbb{E}[\boldsymbol{\theta}_t]$ can and will lie in the interior of $\{\pm 1\}^M$. Accordingly, we will assume the optimization oracle returns a solution $(h^*, \boldsymbol{\theta}^*) \in \mathcal{H} \times \{\pm 1\}^M$ satisfying (3).

### 4.1. Online Multicalibration via Admissibility and Implementability

We now give an algorithm for OLPO that is situated in the *generalized Follow-the-Perturbed-Leader* (GFTPL) framework (Dudík et al., 2020), and describe the conditions under which it can be made efficient with respect to the oracle in Definition 4.1. In particular, our algorithm will ultimately require only a single oracle call per round. The algorithm is roughly constructed in two steps.

The first step is to design an algorithm in the GFTPL framework that obtains sublinear regret. The main idea is to draw a lower-dimensional random vector $\boldsymbol{\alpha} \in \mathbb{R}^N$ where each $\alpha_j$ is drawn independently from an appropriate distribution, and $N \ll |\mathcal{H}|$—essentially, we will think of runtime that is linear in $N$ to be acceptable. The payoff of each of the algorithm's actions is perturbed by some linear combination of $\boldsymbol{\alpha}$ as given by a *perturbation translation matrix* $\boldsymbol{\Gamma} \in [-B, B]^{(|\mathcal{H}| \times 2^M) \times N}$ (note that $|\mathcal{H}| \times 2^M$ denotes the size of the action space of the algorithm). In each round $t \in [T]$, the algorithm selects $(h_t, \boldsymbol{\theta}_t) \in \mathcal{H} \times \{\pm 1\}^M$ to approximately maximize:

$$\max_{h \in \mathcal{H}, \boldsymbol{\theta} \in \{\pm 1\}^M} \left\{ \sum_{s=1}^{t-1} \langle \boldsymbol{\theta}, h(\mathbf{x}_s) \cdot \boldsymbol{f}_s \rangle + \boldsymbol{\alpha} \cdot \boldsymbol{\Gamma}_{(h, \boldsymbol{\theta})} \right\} \quad (4)$$

up to an additive error $\epsilon > 0$. We will show that is equivalent to the oracle (3) in Definition 4.1 (refer to the "implementability condition in Definition 4.3). This algorithm is described in Algorithm 3, and is no-regret up to the error of the oracle $\epsilon$ as long as $\boldsymbol{\Gamma}$ satisfies the following $\delta$-admissibility condition.

**Definition 4.2** ($\delta$-admissibility (Dudík et al., 2020)). A translation matrix $\boldsymbol{\Gamma}$ is $\delta$-admissible if its rows are distinct, and distinct elements within each column differ by at least $\delta$. Formally, in our framework, for every pair $(h, \boldsymbol{\theta}) \neq (h', \boldsymbol{\theta}')$ we need $\boldsymbol{\Gamma}_{(h, \boldsymbol{\theta})} \neq \boldsymbol{\Gamma}_{(h', \boldsymbol{\theta}')}$ and for every $j \in [N]$ we either have $\Gamma_{(h, \boldsymbol{\theta}), j} = \Gamma_{(h', \boldsymbol{\theta}'), j}$ or $|\Gamma_{(h, \boldsymbol{\theta}), j} - \Gamma_{(h', \boldsymbol{\theta}'), j}| \geq \delta$.

It turns out that the aforementioned $\delta$-admissibility condition suffices to design an algorithm that obtains an overall regret guarantee of $O(N\sqrt{T}/\delta + \epsilon T)$. However, this

---

**Algorithm 3** Generalized FTPL for OLPO

1: **Input:** perturbation matrix $\boldsymbol{\Gamma} \in [0, 1]^{(|\mathcal{H}| \times 2^M) \times N}$, and accuracy $\epsilon > 0$
2: randomness $\boldsymbol{\alpha} = (\alpha_1, \cdots, \alpha_N)$: $\alpha_i \sim \text{Unif}[0, \sqrt{T}]$
3: **for** $t = 1, \ldots, T$ **do**
4:     select $(h_t, \boldsymbol{\theta}_t) \in \mathcal{H} \times \{\pm 1\}^M$ per (4)
5:     observe context $\mathbf{x}_t$ and reward vector $\boldsymbol{f}_t$,
6:     receive payoff $\langle \boldsymbol{\theta}_t, h_t(\mathbf{x}_t) \cdot \boldsymbol{f}_t \rangle$
7: **end for**

---

does not yet guarantee oracle-efficiency since a naive construction of $\boldsymbol{\Gamma}$ would require space that is linear in $|\mathcal{H}|$ and exponential in $M$. Thus, we need the property that the perturbations to each action $(h, \boldsymbol{\theta})$ can be simulated efficiently through the optimization oracle (3); that is, without needing to actually access $\boldsymbol{\Gamma}$. This requirement is captured by the *implementability* condition, stated next.

**Definition 4.3** (Implementability). A translation matrix $\boldsymbol{\Gamma}$ is implementable with complexity $D$ if for any realization of the random vector $\boldsymbol{\alpha}$, there exists a set of coefficients $\{\kappa_j\}_{j=1}^{D}$ and vectors $\{\widehat{\boldsymbol{f}}_j\}_{j=1}^{D}$ (both of which will depend on $\boldsymbol{\alpha}$) such that we can express, for all $(h, \boldsymbol{\theta}) \in \mathcal{H} \times \{\pm 1\}^M$,

$$\boldsymbol{\alpha} \cdot \boldsymbol{\Gamma}_{(h, \boldsymbol{\theta})} = \sum_{j=1}^{D} \kappa_j \cdot \left\langle \boldsymbol{\theta}, h(\mathbf{x}_j) \cdot \widehat{\boldsymbol{f}}_j \right\rangle. \quad (5)$$

Crucially, this allows us to simulate (4) by (5) (which is the oracle in Definition 4.1). We note that the definition differs slightly from the definition in Dudík et al. (2020) but they can be shown to be equivalent, and this definition turns out to be slightly more convenient to work with in the proof.

The main theorem of this section shows that the GFTPL framework achieves sublinear regret for OLPO.

**Theorem 4.4.** *Consider the instance of* OLPO *with the decision set* $\mathcal{H} \times \{\pm 1\}^M$. *Suppose that the perturbation translation matrix* $\boldsymbol{\Gamma} \in [-B, B]^{(|\mathcal{H}| \times 2^M) \times N}$ *satisfies* $\delta$-*admissibility and satisfies implementability with complexity* $D$. *Then, Algorithm 3 executed with* $\epsilon = 1/\sqrt{T}$ *satisfies the expected regret* $R_T(\mathcal{L}; \mathcal{H})$ *guarantee*

$$R_T(\mathcal{L}; \mathcal{H}) \leq O\left(\frac{B^2 N \sqrt{T}}{\delta}\right). \quad (6)$$

*Furthermore, this algorithm requires a single oracle call per round, and the per-round complexity is* $O(T + ND)$.

The proof of Theorem 4.4 is provided in Appendix D.1. The key steps are similar to the analysis of the GFTPL algorithm in (Dudík et al., 2020), i.e., the analysis includes characterizing an approximation error term and stability error term. We note that combining (6) with Theorem 2.1 gives a "blackbox" algorithm for obtaining online $\ell_1$-multicalibration given a $\delta$-admissible and implementable $\boldsymbol{\Gamma}$.

## 4.2. Our Admissible and Implementable Constructions

In this section, we give settings in which one can explicitly construct a $\mathbf{\Gamma}$ matrix that is admissible and implementable. Specifically, we will give efficient algorithms for the *transductive* setting and the *small-separator* setting, defined next. We also assume in this section that $\mathcal{H}$ is binary-valued, i.e., $h : \mathcal{X} \to \{0,1\}$ for all $h \in \mathcal{H}$.

**Transductive Setting.** In the transductive setting, the learner has access to the set $\mathcal{X}$ from which contexts will be drawn. Formally, we will say that $\mathcal{X} = \{\mathbf{x}^1, \ldots, \mathbf{x}^D\}$ and that at each round $t$, the context $\mathbf{x}_t \in \mathcal{X}$.

**Small-Separator Setting.** In the small-separator setting, we assume access to a small set of contexts $\mathcal{X}$, called a *separator*. Let $\mathcal{X} = \{\mathbf{x}^1, \ldots, \mathbf{x}^D\}$, and we have that for any two groups $h, h' \in \mathcal{H}$, there exists a feature $\mathbf{x} \in \mathcal{X}$ such that $h(\mathbf{x}) \neq h'(\mathbf{x})$.

These settings are commonly used, even for the simpler problem of oracle-efficient online prediction (Dudík et al., 2020; Syrgkanis et al., 2016), and have since been built on to ensure oracle-efficient online learning under smoothed data or transductive learning with hints[2] (Block et al., 2022; Haghtalab et al., 2022). We are now ready to define our construction of the translation matrix $\mathbf{\Gamma}$ for these two settings. In particular, we prove the following (which proves Theorem 1.7 by combining with Theorem 2.1 and Theorem 4.4).

**Lemma 4.5.** *In the transductive and small-separator settings, there exists a perturbation translation matrix $\mathbf{\Gamma} \in [-B, B]^{(|\mathcal{H}| \times 2^M) \times (DM)}$ that is 1-admissible and implementable with complexity $D$.*

The proof of Lemma 4.5 is deferred to Appendix D.2. We now examine implications of Lemma 4.5 for oracle-efficient multicalibration. Plugging in $\delta = 1$ and $N = DM$ into the statement of Theorem 4.4, and appealing to the reduction in Theorem 2.1 gives us

$$
\begin{aligned}
\mathbb{E}[K(\pi_T, \mathcal{H})] \ \leq \ & \frac{B}{m} + O\left(BDM\sqrt{T}\right) \\
& + 4B\sqrt{\frac{m \log(6T|\mathcal{H}|)}{T}} + \frac{4mB}{T}.
\end{aligned}
$$

Finally, setting the allowable error in the optimization oracle to $\epsilon = \frac{1}{M} = \frac{1}{T^{1/4}}$ and using $M = m + 1$ gives us

$$
\mathbb{E}[K(\pi_T, \mathcal{H}_B)] \leq O\left(BDT^{-1/4}\sqrt{\log(T|\mathcal{H}|)}\right),
$$

which is the bound stated in Section 1.2. In Section 4.2, we give settings where one can construct a $\mathbf{\Gamma}$ matrix that

is admissible and implementable. Specifically, we give efficient algorithms for the *transductive* and *small-separator* settings when $\mathcal{H}$ is binary-valued, i.e., $h : \mathcal{X} \to \{0, 1\}$ for all $h \in \mathcal{H}$, and prove Theorem 1.7.

**Potential improvements and extensions of analysis.** It is natural to ask about the extent to which the assumptions we have made — access to the offline oracle, binary-valued $\mathcal{H}$ and either transductive or sufficiently separated data — can be weakened or improved. We begin by noting that our proofs will directly work given a "$C$-approximate oracle" (3) (to prove $C$-approximate regret, which would suffice for our reduction in Theorem 2.1) since they rely on the GFTPL framework of Dudík et al. (2020) in a black-box manner. Two concurrent recent works (Block et al., 2022; Haghtalab et al., 2022) showed that the generalized FTPL framework can provide oracle-efficient regret bounds for online supervised learning on infinite-sized hypothesis classes with bounded VC dimension (in the case of binary labels) or pseudodimension (in the case of real-valued labels) if the contexts are *smoothed* with respect to some known base probability measure. We believe that adapting their proof technique, especially for the stability term, to the OLPO instance is an interesting direction for future work.

At a more fundamental level, the discrepancy in rates between inefficient and oracle-efficient multicalibration arises solely from the linear dependence on $m$ (or $M$) in the regret bound of oracle-efficient OLPO, as compared to the $O(m^{1/2})$ dependence in the inefficient OLPO implementation. Improving the dependence on $m$ in the oracle-efficient regret bound would require improving the regret analysis of (Dudík et al., 2020), which would be of independent interest.

## Acknowledgments

We are grateful to the anonymous reviewers of ICML for valuable feedback. The last author is thankful to Marco Molinaro for discussions on online linear optimization over general convex bodies and the proof of Lemma 3.3.

## Impact Statement

This paper presents work whose goal is to advance the field of Machine Learning. There are many potential societal consequences of our work, none which we feel must be specifically highlighted here.

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

## A. Further Related Work

**Calibration.** The notion of (sequential) calibration has been extensively studied in statistical learning and forecasting (Dawid, 1982; Foster & Vohra, 1998; Hart, 2022). Dawid (1982) introduced the notion of calibration, and Foster & Vohra (1998) showed the existence of an algorithm capable of producing calibrated forecasts in an online adversarial setting. Subsequently, numerous algorithms were discovered for calibration; see, for example, Foster & Kakade (2006); Fudenberg & Levine (1999); Hart & Mas-Colell (2000); Perchet (2009); Sandroni (2003); Sandroni et al. (2003). Foster (1999) has given a calibration algorithm based on Blackwell approachability, while Abernethy et al. (2011) showed a connection between calibration and no-regret learning (via Blackwell approachability).

**Multicalibration.** The concept of multicalibration extends standard calibration by requiring calibrated predictions not just overall, but across multiple subpopulations defined by a hypothesis class. This notion was first introduced by Hébert-Johnson et al. (2018) in the batch setting. They showed that multicalibrated predictors could provide strong fairness guarantees while maintaining predictive accuracy. Since then, multicalibration and some analogous notions have been studied; for example, Kim et al. (2019) study multiaccuracy, Jung et al. (2021) give algorithms for moment multicalibration, and Gupta et al. (2022) investigates quantile multicalibration. Another line of work explores the connection between multicalibration and *omniprediction* (Garg et al., 2024; Gopalan et al., 2022; 2023). Omniprediction is a paradigm for loss minimization that was introduced in Gopalan et al. (2022). Informally, an omnipredictor is a prediction algorithm that could be used for minimizing a large class of loss functions such that its performance is comparable to some benchmark class of models $\mathcal{F}$. Gopalan et al. (2022) show that we can reduce omniprediction to a $\ell_1$-multicalibration. In particular, if a prediction algorithm is *multicalibrated* (in the $\ell_1$ metric) with respect to some benchmark class of models $\mathcal{H}$, then it is an omnipredictor with respect to all Lipschitz convex losses and the class $\mathcal{H}$. The problem of online omniprediction was introduced in Garg et al. (2024), and they used similar ideas to that of Gopalan et al. (2022) and Globus-Harris et al. (2023) to show that online omniprediction can be reduced to online $\ell_1$-multicalibration. Furthermore, they provided an efficient reduction from online multicalibration to online squared error regression over $\mathcal{H}$, yielding oracle-efficient algorithms for online $\ell_1$-multicalibration and, consequently, for online omniprediction. Other works have also explored online multicalibration under different settings; for example, Gupta et al. (2022) and Lee et al. (2022) provided algorithms that guarantee online multicalibration in the $\ell_\infty$ metric.

**Oracle Efficient Algorithms and Online Multigroup Learning.** No-regret algorithms based on injecting random perturbations, such as Follow-the-Perturbed-Leader, have an early history in lending themselves to computational efficiency on specially structured combinatorial problems, such as the shortest path problem and prediction with decision trees (Kalai & Vempala, 2005). In these problems, the "oracle" constitutes solving a shortest path problem or learning an optimal decison tree from offline data. Motivated by this, Hazan & Koren (2016) posed the more general problem of achieving computational efficiency in *online* learning with respect to an *offline* optimization oracle and showed that this goal is not achievable in the worst case. Subsequently, Block et al. (2022); Daskalakis & Syrgkanis (2016); Dudík et al. (2020); Haghtalab et al. (2022); Syrgkanis et al. (2016); Wang et al. (2022) showed that oracle-efficient learning is possible under further assumptions—both for a variety of combinatorial settings involving market design, and for online supervised learning involving contexts $\{\mathbf{x}_t\}_{t=1}^T$ and labels $\{y_t\}_{t=1}^T$. For the learning settings the results make assumptions on the contexts $\{\mathbf{x}_t\}_{t=1}^T$, but not on the labels. The setting of online $\ell_1$-multicalibration is more reminiscent of (but not exactly the same as) the latter case of online supervised learning. Accordingly, we adopt the assumptions of *transductive* or *sufficiently separated data* made in (Dudík et al., 2020) (which subsume those made in (Daskalakis & Syrgkanis, 2016; Syrgkanis et al., 2016)) and believe our results could be adapted to the weaker assumption of *smoothed data* or *K-hint data* (Block et al., 2022; Haghtalab et al., 2022) in future work. Recently, the oracle-efficient framework was also adopted for online multigroup learning with the aim of minimizing *group-regret* (Acharya et al., 2024; Deng et al., 2024). While group-regret could be closely related to multicalibration and the associated OLPO instance that we set up, it is unclear how to adapt the techniques in (Deng et al., 2024), which are tailored to binary labels and loss functions, to the OLPO problem.

There also exists a rich body of work on oracle efficiency with respect to either *online regression oracles* or *cost-sensitive classification oracles* in the contextual bandits literature (see, e.g. (Agarwal et al., 2014; Foster & Rakhlin, 2020)). Indeed, an online regression oracle was assumed by (Garg et al., 2024) but in a completely different manner from the contextual bandit application. We do not adopt these frameworks for oracles because the contextual bandit problem is different in scope and unnecessary to solve for multicalibration. It is also arguably more difficult than full-information contextual learning: the strongest of the aforementioned results assume stochasticity in the labels $\{y_t\}_{t=1}^T$ and that they are *realized* by a function in the hypothesis class, and still require an online regression oracle, which is stronger than an offline oracle and is only known

to be solvable in the special case of linear models (Azoury & Warmuth, 2001).

## B. Missing Proofs from Section 2

### B.1. Proof of Theorem 2.1

*Proof of Theorem 2.1.* We first reduce online $\ell_1$-calibration error for any group $h$ to the $\ell_1$-norm of the vector $\frac{1}{T}\sum_{t=1}^{T} h(\mathbf{x}_t)\cdot \boldsymbol{f}_t$ plus a small additive "error" term. Recall, the expected online $\ell_1$-multicalibration for group $h \in \mathcal{H}$:

$$\mathbb{E}\left[K(\pi_T, h)\right] = \frac{1}{T}\,\mathbb{E}\left[\sum_{p\in\mathcal{P}}\left|\left(\sum_{t=1}^{T}\mathbb{I}\{p_t=p\}\cdot h(\mathbf{x}_t)\cdot(y_t-p)\right)\right|\right]. \tag{7}$$

We use the following lemma to relate the indicator random variables $\{\mathbb{I}\{p_t=p\}\}_{p\in\mathcal{P}}$ to their corresponding expectations $\{\mathbf{w}_t(p)\}_{p\in\mathcal{P}}$, where $\mathbf{w}_t(p)$ denotes the probability that $p_t$ equals $p$.

**Lemma B.1.** *We have*

$$\mathbb{E}\left[\max_{h\in\mathcal{H}}\left\{\sum_{p\in\mathcal{P}}\left|\sum_{t=1}^{T}\left(\mathbb{I}\{p_t=p\}-\mathbf{w}_t(p)\right)\cdot h(\mathbf{x}_t)\cdot(y_t-p)\right|\right\}\right]\le 4B\sqrt{Tm\log(6T|\mathcal{H}|)}\,+\,4mB.$$

The proof of this lemma relies on a vector version of Azuma–Hoeffding inequality and is deferred to Appendix B.2. We will next use it to complete the proof of Theorem 2.1.

We now proceed to upper bound the online multicalibration error for $\mathcal{H}$. Using (7) along with the triangle inequality, we get

$$\mathbb{E}\left[K(\pi_T, \mathcal{H})\right] = \frac{1}{T}\,\mathbb{E}\left[\max_{h\in\mathcal{H}}\left\{\sum_{p\in\mathcal{P}}\left|\sum_{t=1}^{T}\mathbb{I}\{p_t=p\}\cdot h(\mathbf{x}_t)\cdot(y_t-p)\right|\right\}\right]$$

$$\le \frac{1}{T}\,\max_{h\in\mathcal{H}}\left\{\sum_{p\in\mathcal{P}}\left|\sum_{t=1}^{T}\mathbf{w}_t(p)\cdot h(\mathbf{x}_t)\cdot(y_t-p)\right|\right\}\,+$$

$$\frac{1}{T}\,\mathbb{E}\left[\max_{h\in\mathcal{H}}\left\{\sum_{p\in\mathcal{P}}\left|\sum_{t=1}^{T}(\mathbb{I}\{p_t=p\}-\mathbf{w}_t(p))\cdot h(\mathbf{x}_t)\cdot(y_t-p)\right|\right\}\right]$$

$$\le \frac{1}{T}\,\max_{h\in\mathcal{H}}\left\{\left\|\sum_{t=1}^{T}h(\mathbf{x}_t)\cdot \boldsymbol{f}_t\right\|_1\right\}+4B\sqrt{\frac{m\log(6T|\mathcal{H}|)}{T}}+\frac{4mB}{T}, \tag{8}$$

where the last inequality follows from Lemma B.1 together with the definition of $\boldsymbol{f}_t$ in (2). Now using the definition of the dual norm, we can write

$$\frac{1}{T}\,\max_{h\in\mathcal{H}}\left\|\sum_{t=1}^{T}h(\mathbf{x}_t)\cdot \boldsymbol{f}_t\right\|_1 = \frac{1}{T}\,\max_{h\in\mathcal{H},\boldsymbol{\theta}\in B_\infty}\left\langle\boldsymbol{\theta}, \sum_{t=1}^{T}h(\mathbf{x}_t)\cdot \boldsymbol{f}_t\right\rangle$$

$$\le \frac{1}{T}\,\sum_{t=1}^{T}\langle\boldsymbol{\theta}_t, h_t(\mathbf{x}_t)\cdot \boldsymbol{f}_t\rangle+\frac{R_T(\mathcal{L};\mathcal{H})}{T}\le \frac{B}{m}+\frac{R_T(\mathcal{L};\mathcal{H})}{T}, \tag{9}$$

where the first inequality follows by applying the regret bound for OLPO obtained by $\mathcal{L}$ and the second inequality follows from the definition of the halfspace oracle (see Definition 2.2). We note that we can replace the "sup" operator with a "max" operator in the equality above by the compactness of $B_\infty = \{\boldsymbol{\theta} : \|\boldsymbol{\theta}\|_\infty \le 1\}$ and the continuity of the linear function. Combining (8) and (9) completes the proof of Theorem 2.1. $\qquad\square$

### B.2. Proof of Lemma B.1

The proof relies on a "vector" form of the Azuma-Hoeffding inequality (see Theorem 1.8 in (Hayes, 2005)).

**Theorem B.2** (Vector Azuma–Hoeffding's inequality). *Let* $\mathbf{S}_n = \sum_{t=1}^{n} \mathbf{X}_t$ *be a martingale relative to the sequence* $\mathbf{X}_1, \ldots, \mathbf{X}_n$ *where each* $\mathbf{X}_t$ *takes values in* $\mathbb{R}^d$ *and satisfies (i)* $\mathbb{E}[\mathbf{X}_t] = \mathbf{0}$ *and (ii)* $\|\mathbf{X}_t\|_2 \leq c$. *Then, for any* $\eta > 0$ *and* $n \geq 1$, *we have*

$$\mathbf{P}(\|\mathbf{S}_n\|_2 \geq \eta) \leq 2e^2 \exp\left(\frac{-\eta^2}{2nc^2}\right).$$

Given this inequality, we can now complete the proof of Lemma B.1.

*Proof of Lemma B.1.* For each $h \in \mathcal{H}$ and $t \in [T]$, we define the vector $\mathbf{Y}_t^h \in \mathbb{R}^M$ such that

$$\mathbf{Y}_t^h(p) = \left(\mathbb{I}\{p_t = p\} - \mathbf{w}_t(p)\right) \cdot h(\mathbf{x}_t) \cdot (y_t - p) .$$

We now verify that the conditions on the sequence $\{\mathbf{Y}_t^h(p)$ for the Vector Azuma-Hoeffding inequality (Theorem B.2) hold. We note that

$$\|\mathbf{Y}_t^h\|_2 = \sqrt{\sum_{p \in \mathcal{P}} \mathbf{Y}_t^h(p)^2} = \sqrt{\sum_{p \in \mathcal{P}} h(\mathbf{x}_t)^2 \cdot (y_t - p)^2 \cdot (\mathbb{I}\{p_t = p\} - \mathbf{w}_t(p))^2}$$

$$\leq \max_{\mathbf{x} \in \mathcal{X}} |h(\mathbf{x})| \cdot \sqrt{\sum_{p \in \mathcal{P}} (\mathbb{I}\{p_t = p\} - \mathbf{w}_t(p))^2} \leq \sqrt{2}B,$$

where the final inequality follows from the fact that $h \in \mathcal{H} \subseteq \mathcal{H}_B$, $\sum_{p \in \mathcal{P}} \mathbb{I}\{p_t = p\} = 1$, and that $\mathbf{w}_t$ is a distribution. This verifies condition (ii). Next, observe that

$$\mathbb{E}[\mathbf{Y}_t^h(p) \mid \mathbf{Y}_1^h, \ldots, \mathbf{Y}_{t-1}^h] = \mathbb{E}[h(\mathbf{x}_t) \cdot (y_t - p) \cdot (\mathbb{I}\{p_t = p\} - \mathbf{w}_t(p)) \mid \mathbf{Y}_1^h, \ldots, \mathbf{Y}_{t-1}^h]$$

$$= h(\mathbf{x}_t) \cdot (y_t - p) \cdot \mathbb{E}[(\mathbb{I}\{p_t = p\} - \mathbf{w}_t(p)) \mid \mathbf{Y}_1^h, \ldots, \mathbf{Y}_{t-1}^h] = 0$$

since $\mathbf{w}_t$ only depends on information from rounds $1, \ldots, t-1$, and $\mathbb{E}[(\mathbb{I}\{p_t = p\}) \mid \mathbf{Y}_1^h, \ldots, \mathbf{Y}_{t-1}^h] = \mathbf{w}_t(p)$. This verifies condition (i). Thus, $\mathbf{S}_n^h = \sum_{t=1}^{n} \mathbf{Y}_t^h$ is a martingale with respect to $\mathbf{Y}_1^h, \mathbf{Y}_2^h, \ldots$, and

$$\|\mathbf{S}_T^h\|_1 = \sum_{p \in \mathcal{P}} \left| \left(\sum_{t=1}^{T} \mathbb{I}\{p_t = p\} \cdot h(\mathbf{x}_t) \cdot (y_t - p)\right) - \left(\sum_{t=1}^{T} \mathbf{w}_t(p) \cdot h(\mathbf{x}_t) \cdot (y_t - p)\right) \right|.$$

Furthermore, since $\mathbf{S}_T^h \in \mathbb{R}^M$, we have $\|\mathbf{S}_T^h\|_1 \leq \sqrt{M} \cdot \|\mathbf{S}_T^h\|_2$, and applying Theorem B.2 implies that

$$\mathbf{P}\left(\|\mathbf{S}_T^h\|_2 \geq 2\sqrt{2}B\sqrt{T \log(6T|\mathcal{H}|)}\right) \leq 2e^2 \exp\left(\frac{-8B^2 T \log(6T|\mathcal{H}|)}{4B^2 T}\right) = \frac{2e^2}{36(T)^2|\mathcal{H}|^2} \leq \frac{1}{T|\mathcal{H}|}.$$

Thus, with probability at least $1 - 1/(T|\mathcal{H}|)$, we have

$$\|\mathbf{S}_T^h\|_1 \leq \sqrt{M} \cdot \|\mathbf{S}_T^h\|_2 \leq \sqrt{2m} \cdot 2\sqrt{2}B\sqrt{T \log(6T|\mathcal{H}|)} \leq 4B\sqrt{Tm \log(6T|\mathcal{H}|)}, \tag{10}$$

since $M = m + 1 \leq 2m$. We are now ready to complete the proof. Let $G_h$ denote the event that $\|\mathbf{S}_T^h\|_1 \leq 4B\sqrt{Tm \log(6T|\mathcal{H}|)}$ for $h \in \mathcal{H}$. Let $G$ denote the event that $G_h$ holds *simultaneously* for all $h \in \mathcal{H}$; i.e. $G = \cap_{h \in \mathcal{H}} G_h$. By the union bound over all $h \in \mathcal{H}$, we can conclude that $\mathbf{P}(G) \geq 1 - 1/T$. Then, we have

$$\mathbb{E}\left[\max_{h \in \mathcal{H}} \{\|\mathbf{S}_T^h\|_1\}\right] = \mathbb{E}\left[\max_{h \in \mathcal{H}} \{\|\mathbf{S}_T^h\|_1\} \mid G\right] \cdot \mathbf{P}(G) + \mathbb{E}\left[\max_{h \in \mathcal{H}} \{\|\mathbf{S}_T^h\|_1\} \mid \overline{G}\right] \cdot \mathbf{P}(\overline{G})$$

$$\leq 4B\sqrt{Tm \log(6T|\mathcal{H}|)} \cdot \left(1 - \frac{1}{T}\right) + \mathbb{E}\left[\max_{h \in \mathcal{H}} \{\|\mathbf{S}_T^h\|_1\} \mid \overline{G}\right] \cdot \mathbf{P}(\overline{G})$$

$$\leq 4B\sqrt{Tm \log(6T|\mathcal{H}|)} + 2MBT \cdot \frac{1}{T} \leq 4B\sqrt{Tm \log(6T|\mathcal{H}|)} + 4mB.$$

where the first inequality uses (10) for all $h \in \mathcal{H}$ and the second inequality bounds $\|\mathbf{S}_T^h\|_1 \leq 2MTB$ for all $h \in \mathcal{H}$. The final inequality uses $M = m + 1 \leq 2m$. This completes the proof of the lemma. $\square$

## B.3. Constructing the Halfspace Oracle

Here we repeat the construction of an efficient halfspace oracle from (Abernethy et al., 2011) for our setting. In particular, we will design an efficient oracle $\mathcal{O}$ that can, given $\mathbf{x} \in \mathcal{X}$, $h \in \mathcal{H}$ and $\boldsymbol{\theta} \in B_\infty$, select a distribution $\mathbf{w} \in \mathbb{R}^M$ such that for all $y \in [0,1]$, we have

$$\sum_{i=0}^{m} \boldsymbol{\theta}(i) \cdot h(\mathbf{x}) \cdot \mathbf{w}(i) \cdot \left( y - \frac{i}{m} \right) \leq \frac{B}{m}.$$

We describe this oracle in Algorithm 4.

---

**Algorithm 4** $\mathcal{O}(\mathbf{x}, h, \boldsymbol{\theta})$

1: **Input:** $\mathbf{x} \in \mathcal{X}$, $h \in \mathcal{H} \subseteq \mathcal{H}_B$ and $\boldsymbol{\theta} : ||\boldsymbol{\theta}||_\infty \leq 1$
2: define $\widehat{\boldsymbol{\theta}} : \widehat{\boldsymbol{\theta}}(i) = \boldsymbol{\theta}(i) \cdot h(\mathbf{x})$
3: **if** $\widehat{\boldsymbol{\theta}}(0) \leq 0$ **then**
4:     $\mathbf{w} \leftarrow \delta_0$
5: **else if** $\widehat{\boldsymbol{\theta}}(m) \geq 0$ **then**
6:     $\mathbf{w} \leftarrow \delta_m$
7: **else**
8:     find coordinate $i$ such that $\widehat{\boldsymbol{\theta}}(i) > 0$ and $\widehat{\boldsymbol{\theta}}(i+1) \leq 0$
9:     $\mathbf{w} \leftarrow \frac{\widehat{\boldsymbol{\theta}}(i+1)}{\widehat{\boldsymbol{\theta}}(i+1) - \widehat{\boldsymbol{\theta}}(i)} \cdot \delta_i + \frac{\widehat{\boldsymbol{\theta}}(i)}{\widehat{\boldsymbol{\theta}}(i) - \widehat{\boldsymbol{\theta}}(i+1)} \cdot \delta_{i+1}$
10: **end if**
11: **return** $\mathbf{w}$

---

It immediately follows from the description that $\mathbf{w}$ is a valid distribution. Furthermore, note that this oracle can be implemented efficiently. If $\widehat{\boldsymbol{\theta}}(0) > 0$ and $\widehat{\boldsymbol{\theta}}(m) < 0$, then there must exist $i$ such that $\widehat{\boldsymbol{\theta}}(i) > 0$ and $\widehat{\boldsymbol{\theta}}(i+1) \leq 0$, and such an index can be found using binary search. Thus, this algorithm requires at most $O(\log(m))$ computations. The following lemma proves the main property of the halfspace oracle.

**Lemma B.3.** *Given any $\mathbf{x} \in \mathcal{X}$, $h \in \mathcal{H}$ and $\boldsymbol{\theta} \in B_\infty$, let $\mathbf{w} = \mathcal{O}(\mathbf{x}, h, \boldsymbol{\theta})$ be the output of Algorithm 4. Then, for any $y \in [0,1]$, we have $\sum_{i=0}^{m} \boldsymbol{\theta}(i) \cdot h(\mathbf{x}) \cdot \left( y - \frac{i}{m} \right) \cdot \mathbf{w}(i) \leq \frac{B}{m}$.*

*Proof.* We first note that we can write

$$\sum_{i=0}^{m} \boldsymbol{\theta}(i) \cdot h(\mathbf{x}) \cdot \left( y - \frac{i}{m} \right) \cdot \mathbf{w}(i) = \sum_{i=0}^{m} \widehat{\boldsymbol{\theta}}(i) \cdot \left( y - \frac{i}{m} \right) \cdot \mathbf{w}(i),$$

where we have used $\widehat{\boldsymbol{\theta}}(i) = \boldsymbol{\theta}(i) \cdot h(\mathbf{x})$ for all $i = 0, 1, \ldots, m$. Observe that if $\widehat{\boldsymbol{\theta}}(0) \leq 0$ or $\widehat{\boldsymbol{\theta}}(m) \geq 0$, the lemma is trivially true. Otherwise, we have

$$\sum_{i=0}^{m} \widehat{\boldsymbol{\theta}}(i) \cdot \left( y - \frac{i}{m} \right) \cdot \mathbf{w}(i) = \left( y - \frac{i}{m} \right) \cdot \mathbf{w}(i) \cdot \widehat{\boldsymbol{\theta}}(i) + \left( y - \frac{i+1}{m} \right) \cdot \mathbf{w}(i+1) \cdot \widehat{\boldsymbol{\theta}}(i+1)$$

$$= \left( y - \frac{i}{m} \right) \cdot \frac{\widehat{\boldsymbol{\theta}}(i+1)}{\widehat{\boldsymbol{\theta}}(i+1) - \widehat{\boldsymbol{\theta}}(i)} \cdot \widehat{\boldsymbol{\theta}}(i) + \left( y - \frac{i+1}{m} \right) \cdot \frac{\widehat{\boldsymbol{\theta}}(i)}{\widehat{\boldsymbol{\theta}}(i) - \widehat{\boldsymbol{\theta}}(i+1)} \cdot \widehat{\boldsymbol{\theta}}(i+1)$$

$$= -\frac{1}{m} \cdot \frac{\widehat{\boldsymbol{\theta}}(i) \cdot \widehat{\boldsymbol{\theta}}(i+1)}{\widehat{\boldsymbol{\theta}}(i) - \widehat{\boldsymbol{\theta}}(i+1)} \leq \frac{1}{m} \cdot \frac{\max\{|\widehat{\boldsymbol{\theta}}(i)|, |\widehat{\boldsymbol{\theta}}(i+1)|\}}{2} \leq \frac{B}{m},$$

where the penultimate inequality uses the AM-HM inequality, and the final inequality follows from the fact that $\boldsymbol{\theta} \in B_\infty$ and $\max_{\mathbf{x} \in \mathcal{X}} |h(\mathbf{x})| \leq B$ for all $h \in \mathcal{H}$. $\qquad\square$

## C. Missing Proofs from Section 3

### C.1. Proof of Lemma 3.2.

*Proof.* We begin by describing how to obtain a learning algorithm $\mathcal{L}$ for OLPO using a learning algorithm $\widetilde{\mathcal{L}}$ for Lin-OLPO. To achieve this, two components are required: (1) translating decisions from $\widetilde{\mathcal{L}}$ to $\mathcal{L}$, and (2) generating reward vectors for the Lin-OLPO instance using rewards from the OLPO instance. For each round $t \in [T]$, given decision $\widetilde{\boldsymbol{\theta}}_t$ from $\widetilde{\mathcal{L}}$, set $h_t = h$ with probability $\gamma_t(h) = \|\widetilde{\boldsymbol{\theta}}_t(h)\|_\infty$, and $\boldsymbol{\theta}_t = (1/\gamma_t(h)) \cdot \widetilde{\boldsymbol{\theta}}_t(h_t) \in \mathbb{B}_\infty$. Then, $(h_t, \boldsymbol{\theta}_t)$ corresponds to the decision taken by $\mathcal{L}$ for the OLPO instance. Next, given context $\mathbf{x}_t$ and reward vector $\boldsymbol{f}_t$ from the OLPO instance, let $\widehat{\boldsymbol{f}}_t = \frac{1}{B \cdot L} \cdot (h^{(1)}(\mathbf{x}_t) \cdot \boldsymbol{f}_t, \ldots, h^{(|\mathcal{H}|)} \cdot \boldsymbol{f}_t)$ be the reward vector used to simulate the Lin-OLPO instance. Note that $\widehat{\boldsymbol{f}}_t \in \mathbb{B}_{\infty,1}$ since $h \in \mathcal{H}_B$ and $\|\boldsymbol{f}_t\|_1 \leq L$. This completes the reduction from OLPO to Lin-OLPO. Next, we analyze the corresponding regret.

First, we observe that

$$\max_{\widetilde{\boldsymbol{\theta}} \in \mathbb{B}_{1,\infty}} \left\langle \widetilde{\boldsymbol{\theta}}, \sum_{t=1}^T \widetilde{\boldsymbol{f}}_t \right\rangle = \max_{h \in \mathcal{H}} \| \sum_{t=1}^T \widetilde{\boldsymbol{f}}_t(h) \|_1 = \max_{h \in \mathcal{H}} \left\| \sum_{t=1}^T \frac{h(\mathbf{x}_t)}{B \cdot L} \cdot \boldsymbol{f}_t \right\|_1$$

$$= \frac{1}{B \cdot L} \left( \max_{h \in \mathcal{H}, \boldsymbol{\theta} \in \mathbb{B}_\infty} \left\langle \boldsymbol{\theta}, \sum_{t=1}^T h(\mathbf{x}_t) \cdot \boldsymbol{f}_t \right\rangle \right). \tag{11}$$

where in the final equality we used the definition of the dual norm. Furthermore, observe that for any $\widetilde{\boldsymbol{\theta}}_t \in \mathbb{B}_{1,\infty}$ and $\widetilde{\boldsymbol{f}}_t = \frac{1}{B \cdot L}(h^{(1)}(\mathbf{x}_t) \cdot \boldsymbol{f}_t, \cdots, h^{(|\mathcal{H}|)}(\mathbf{x}_t) \cdot \boldsymbol{f}_t)$, we have

$$\mathbb{E}[\langle \boldsymbol{\theta}_t, h_t(\mathbf{x}_t) \cdot \boldsymbol{f}_t \rangle] = B \cdot L \sum_{h \in \mathcal{H}} \gamma_t(h) \cdot \left\langle \frac{\widetilde{\boldsymbol{\theta}}_t(h)}{\gamma_t(h)}, \frac{h(\mathbf{x}_t)}{B \cdot L} \cdot \boldsymbol{f}_t \right\rangle$$

$$= B \cdot L \sum_{h \in \mathcal{H}} \left\langle \widetilde{\boldsymbol{\theta}}_t(h), \widetilde{\boldsymbol{f}}_t(h) \right\rangle = B \cdot L \left\langle \widetilde{\boldsymbol{\theta}}_t, \widetilde{\boldsymbol{f}}_t \right\rangle. \tag{12}$$

where $\gamma_t(h) = \|\widetilde{\boldsymbol{\theta}}_t(h)\|_\infty$ denotes the probability of selecting group $h$ in round $t$ and $\boldsymbol{\theta}_t = (1/\gamma_t(h)) \cdot \widetilde{\boldsymbol{\theta}}_t(h_t)$. On combining (11) and (12), and taking expectations, we can conclude that:

$$R_T(\mathcal{L}; \mathcal{H}, \boldsymbol{\Theta}) = \max_{h \in \mathcal{H}, \boldsymbol{\theta} \in \mathbb{B}_\infty} \left\langle \boldsymbol{\theta}, \sum_{t=1}^T h(\mathbf{x}_t) \cdot \boldsymbol{f}_t \right\rangle - \sum_{t=1}^T \mathbb{E}[\langle \boldsymbol{\theta}_t, h_t(\mathbf{x}_t) \cdot \boldsymbol{f}_t \rangle]$$

$$= B \cdot L \left( \max_{\widetilde{\boldsymbol{\theta}} \in \mathbb{B}_{1,\infty}} \left\langle \widetilde{\boldsymbol{\theta}}, \sum_{t=1}^T \widetilde{\boldsymbol{f}}_t \right\rangle - \sum_{t=1}^T \mathbb{E}\left[ \left\langle \widetilde{\boldsymbol{\theta}}_t, \widetilde{\boldsymbol{f}}_t \right\rangle \right] \right) = B \cdot L \cdot R_T^{\text{Lin-OLPO}}(\widetilde{\mathcal{L}}; \mathbb{B}_{1,\infty}). \tag{13}$$

This completes the proof of the lemma. $\qquad \square$

### C.2. Proof of Lemma 3.3

*Proof.* To bound the regret, we observe

$$\sum_{t=1}^T \mathbb{E}\left[ \left\langle \widetilde{\boldsymbol{\theta}}_t, \widetilde{\boldsymbol{f}}_t \right\rangle \right] = \sum_{t=1}^T \sum_{h \in \mathcal{H}} \gamma_t(h) \cdot \left\langle \boldsymbol{\theta}_t^h, \widetilde{\boldsymbol{f}}_t(h) \right\rangle \geq \max_{h \in \mathcal{H}} \left\{ \sum_{t=1}^T \left\langle \boldsymbol{\theta}_t^h, \widetilde{\boldsymbol{f}}_t(h) \right\rangle \right\} - \sqrt{T \log |\mathcal{H}|}, \tag{14}$$

where the inequality follows by applying the no-regret property of the multiplicative weights update algorithm $\mathcal{E}$ and noting that $\langle \boldsymbol{\theta}_t^h, \widetilde{\boldsymbol{f}}_t(h) \rangle \in [-1, 1]$ for all $h \in \mathcal{H}$ (since $\widetilde{\boldsymbol{f}}_t \in \mathbb{B}_{\infty,1}$). Subsequently, we apply the no-regret property of each of the online gradient descent algorithms $\mathcal{A}_1, \ldots, \mathcal{A}_{|\mathcal{H}|}$ to obtain

$$\sum_{t=1}^T \left\langle \boldsymbol{\theta}_t^h, \widetilde{\boldsymbol{f}}_t(h) \right\rangle \geq \max_{\boldsymbol{\theta} \in B_\infty} \left\{ \sum_{t=1}^T \left\langle \boldsymbol{\theta}, \widetilde{\boldsymbol{f}}_t(h) \right\rangle \right\} - \sqrt{TM} = \|\sum_{t=1}^T \widetilde{\boldsymbol{f}}_t(h)\|_1 - \sqrt{TM}, \tag{15}$$

where the regret bound uses the fact that $\|\widetilde{\boldsymbol{f}}_t(h)\|_2 \leq \|\widetilde{\boldsymbol{f}}_t(h)\|_1 \leq 1$ and that the $\ell_2$ diameter of $\mathbb{B}_\infty$ is $\sqrt{M}$. Combining (14) and (15) gives

$$\sum_{t=1}^{T} \mathbb{E}\left[\left\langle \widetilde{\boldsymbol{\theta}}_t, \widetilde{\boldsymbol{f}}_t \right\rangle\right] \geq \max_{h \in \mathcal{H}} \|\sum_{t=1}^{T} \widetilde{\boldsymbol{f}}_t(h)\|_1 - \sqrt{T \log|\mathcal{H}|} - \sqrt{TM},$$

which upon re-arranging proves the lemma. $\qquad\square$

### C.3. Proof of Lemma 3.4

*Proof.* Fix a transcript $\pi_T = \{(\mathbf{x}_t, p_t, y_t)\}_{t=1}^T$. To avoid notational clutter, we will drop the argument $\pi_T$ in the remainder of the proof. We will also index predictions as $p \in \mathcal{P}$, and use $S(p)$ to denote the set of rounds in which the prediction was $p$; that is, $S(p) = \{t \in [T] : p_t = p\}$. For $h \in \mathcal{H}$, recall

$$K(h) = K(\pi_T, h) = \frac{1}{T} \sum_{p \in \mathcal{P}} \left| \sum_{t \in S(p)} h(\mathbf{x}_t) \cdot (y_t - p) \right|.$$

Furthermore, define $K_p(h) = \left| \sum_{t \in S(p)} h(\mathbf{x}_t) \cdot (y_t - p) \right|$ for every $p \in \mathcal{P}$.

**Claim C.1.** For any $p \in \mathcal{P}$, let $T_p = |S(p)|$. Then, we have for any $h, h' \in \mathcal{H}$:

$$|K_p(h) - K_p(h')| \leq T_p \cdot \|h - h'\|_{\ell_\infty}.$$

Using this claim, we can complete the proof of Lemma 3.4. First, we note that $K(h) = \frac{1}{T} \sum_{p \in \mathcal{P}} K_p(h)$. Since $K(h) \geq 0$, we have $K(h) = |K(h)|$. Using this, and the triangle inequality twice, we get

$$
\begin{aligned}
K(h) - K(h') &= |K(h)| - |K(h')| \leq |K(h) - K(h')| \\
&= \left| \frac{1}{T} \sum_{p \in \mathcal{P}} (K_p(h) - K_p(h')) \right| \leq \frac{1}{T} \sum_{p \in \mathcal{P}} |K_p(h) - K_p(h')| \\
&\leq \frac{1}{T} \sum_{p \in \mathcal{P}} T_p \cdot \|h - h'\|_{\ell_\infty} = \|h - h'\|_{\ell_\infty},
\end{aligned}
$$

where the final inequality uses Claim C.1 and the final equality used $\sum_{p \in \mathcal{P}} T_p = T$. By symmetry, we similarly have

$$K(h') - K(h) \leq \|h - h'\|_{\ell_\infty},$$

from which we conclude that $|K(h) - K(h')| \leq \|h - h'\|_{\ell_\infty}$. Now, fix an $h \in \mathcal{H}$, and pick $h' \in \mathcal{H}_\beta$ such that $\|h - h'\|_{L_\infty} \leq \beta$ (note that such an $h' \in \mathcal{H}_\beta$ always exists according to Definition 1.3). Thus, we have

$$K(h) \leq K(h') + \|h - h'\|_{\ell_\infty} \leq \alpha + \beta,$$

as desired. We complete the proof of Lemma 3.4 by proving Claim C.1.

*Proof of Claim C.1.* Applying the reverse triangle inequality, we have

$$
\begin{aligned}
K(h) - K(h') &= \left| \sum_{t \in S(p)} h(\mathbf{x}_t) \cdot (y_t - p) \right| - \left| \sum_{t \in S(p)} h'(\mathbf{x}_t) \cdot (y_t - p) \right| \\
&\leq \left| \sum_{t \in S(p)} h(\mathbf{x}_t) \cdot (y_t - p) - \sum_{t \in S(p)} h'(\mathbf{x}_t) \cdot (y_t - p) \right| \\
&= \left| \sum_{t \in S(p)} (h(\mathbf{x}_t) - h'(\mathbf{x}_t)) \cdot (y_t - p) \right|.
\end{aligned}
$$

Applying again the triangle inequality to the expression above, and using the fact that $y_t \in [0, 1]$ for all $t \in [T]$, we get

$$K(h) - K(h') \leq \sum_{t \in S(p)} |(h(\mathbf{x}_t) - h'(\mathbf{x}_t))| \leq T_p \cdot \max_{\mathbf{x} \in \mathcal{X}} |h(\mathbf{x}) - h'(\mathbf{x})| = T_p \cdot \|h - h'\|_{\ell_\infty}. \tag{16}$$

By symmetry, we also have the following.

$$K(h') - K(h) \leq T_p \cdot \|h' - h\|_{L_\infty}. \tag{17}$$

Combining (16) and (17) completes the proof. □

This completes the proof of Lemma 3.4. □

### C.4. Applications of Theorem 1.4

In this section, we give applications of Theorem 1.4 to specific function classes.

**Application 1: Polynomial Regression.** Suppose $\mathcal{X} = [0, 1]^d$, and our hypothesis class corresponds to multivariate polynomial regression functions of degree $k$; that is, given $x \in \mathcal{X}$, $h(x) = \sum_{i=1}^{d} \sum_{a=1}^{k} h_{i+(a-1)k} \cdot x_i^a$ where $g \in \mathbb{R}^{kd}$. Suppose that $\mathcal{H}(d, k) = \{h \in \mathbb{R}^{kd} : \|h\|_1 \leq B\}$. Then, constructing an $\beta$-cover for $\mathcal{H}$ in the traditional sense gives us an $\beta$-cover for $\mathcal{H}$ in the functional sense. In particular, since $\mathcal{X} = [0, 1]^d$, we have

$$\max_{x \in \mathcal{X}} |h(x) - h'(x)| \leq \|h - h'\|_1.$$

It is then a standard fact that the $\beta$-covering number of the $\ell_1$ ball of radius $B$ with respect to the $\ell_1$-norm is at most $\left(1 + \frac{2B}{\beta}\right)^{kd}$. Therefore, applying Theorem 1.4 and setting $\beta := T^{-1/3}$ gives us multicalibration error

$$\mathbb{E}[K(\pi_T, \mathcal{H}_{\mathsf{poly}(d,k)})] = \mathcal{O}\left(B(kd)^{1/2} T^{-1/3} \log(BT)\right).$$

The class of bounded linear functions is subsumed by this class (when $k = 1$), for which we obtain the Corollary 1.5.

**Application 2: Bounded, uniformly Lipschitz, convex functions.** Let $\mathcal{H}([0, 1]^d, B, L) \subseteq \mathcal{H}_B$ denote the set of real-valued convex functions defined on $\mathcal{X} := [0, 1]^d$ that are uniformly bounded by $B$ and uniformly Lipschitz[3] with constant $L$. (**?**)Theorem 6]bronshtein1976varepsilon shows that in this case, $|\mathcal{H}_\beta| \leq C\left(\frac{1}{\beta}\right)^{d/2}$, where $C$ is a positive constant that depends only on the Lipschitz parameter $L$. Therefore, applying Theorem 1.4 and setting $\beta := T^{-\frac{1}{2+d/2}}$ yields multicalibration error

$$K(\pi_T, \mathcal{H}([0, 1]^d, B, L)) = \widetilde{\mathcal{O}}(Bd^{1/2} T^{-1/3} \log(BT)) + \widetilde{\mathcal{O}}(T^{-\frac{1}{2+d/2}}).$$

The second term dominates if $d > 2$, and reflects the standard "curse of dimensionality" that we encounter in statistical learning of real-valued convex functions.

## D. Missing Proofs from Section 4

### D.1. Proof of Theorem 4.4

We first show that if the perturbation translation matrix $\mathbf{\Gamma} \in [-B, B]^{(|\mathcal{H}| \times 2^M) \times N}$ satisfies $\delta$-admissibility, then Algorithm 3 has regret bounded by $O(BN\sqrt{T}/\delta)$. Then, we show that since $\Gamma$ satisfies implementability with complexity $D$, we can indeed execute Algorithm 3 efficiently. For ease of exposition, we will use $R_T$ to denote the regret of Algorithm 3. To begin, recall that we defined

$$R_T = \max_{h^* \in \mathcal{H}, \boldsymbol{\theta}^* \in \{\pm 1\}^M} \left\{ \left\langle \boldsymbol{\theta}^*, \sum_{t=1}^{T} h^*(\mathbf{x}_t) \cdot \boldsymbol{f}_t \right\rangle \right\} - \sum_{t=1}^{T} \mathbb{E}[\langle \boldsymbol{\theta}_t, h_t(\mathbf{x}_t) \cdot \boldsymbol{f}_t \rangle]. \tag{18}$$

---

[3]Without the Lipschitz assumption, (Guntuboyina & Sen, 2012) showed that the covering number of is actually infinity.

It suffices to uniformly upper bound the regret incurred by each $h \in \mathcal{H}$; we will provide a worst-case analysis of this quantity for arbitrary reward vectors $\{\boldsymbol{f}_t\}_{t=1}^T$ Formally, we can write

$$R_T = \max_{h \in \mathcal{H}} R_T(h) \text{ where}$$

$$R_T(h) := \max_{\boldsymbol{\theta} \in \{\pm 1\}^M} \left\{ \sum_{t=1}^T \langle \boldsymbol{\theta}, h(\mathbf{x}_t) \cdot \boldsymbol{f}_t \rangle \right\} - \mathbb{E}\left[ \sum_{t=1}^T \langle \boldsymbol{\theta}_t, h_t(\mathbf{x}_t) \cdot \boldsymbol{f}_t \rangle \right].$$

For each $h \in \mathcal{H}$, let $\boldsymbol{\theta}_h^* \in \{\pm 1\}^M$ be a maximizer of the function $\sum_{t=1}^T \langle \boldsymbol{\theta}, h(\mathbf{x}_t) \cdot \boldsymbol{f}_t \rangle$ in the variable $\boldsymbol{\theta}$. Then, we can decompose $R_T(h)$ into two terms as below:

$$R_T(h) = \mathbb{E}\left[ \sum_{t=1}^T \langle \boldsymbol{\theta}_h^*, h(\mathbf{x}_t) \cdot \boldsymbol{f}_t \rangle - \sum_{t=1}^T \langle \boldsymbol{\theta}_t, h_t(\mathbf{x}_t) \cdot \boldsymbol{f}_t \rangle \right]$$

$$= \mathbb{E}\Big[ \underbrace{\sum_{t=1}^T \langle \boldsymbol{\theta}_h^*, h(\mathbf{x}_t) \cdot \boldsymbol{f}_t \rangle - \sum_{t=1}^T \langle \boldsymbol{\theta}_{t+1}, h_{t+1}(\mathbf{x}_t) \cdot \boldsymbol{f}_t \rangle}_{T_1} \Big]$$

$$+ \mathbb{E}\Big[ \underbrace{\sum_{t=1}^T \langle \boldsymbol{\theta}_{t+1}, h_{t+1}(\mathbf{x}_t) \cdot \boldsymbol{f}_t \rangle - \sum_{t=1}^T \langle \boldsymbol{\theta}_t, h_t(\mathbf{x}_t) \cdot \boldsymbol{f}_t \rangle}_{T_2} \Big] \tag{19}$$

The first term $T_1$ corresponds to an *approximation error term*: suppose we could use the clairvoyant decision $(h_{t+1}, \boldsymbol{\theta}_{t+1})$ for round $t$ — without noise, this would be optimal, but with noise it will create extra error. The following lemma(Lemma B.1 in (Dudík et al., 2020)), characterizes $T_1$ pointwise for every realization of the noise $\boldsymbol{\alpha}$.

**Lemma D.1.** *(Be-the-Approximate Leader Lemma) In the Generalized FTPL algorithm, we have*

$$T_1 \leq \boldsymbol{\alpha} \cdot (\boldsymbol{\Gamma}_{(h_1, \boldsymbol{\theta}_1)} - \boldsymbol{\Gamma}_{(h, \boldsymbol{\theta})}) + \epsilon \cdot (T + 1)$$

*for each $(h, \boldsymbol{\theta}) \in \mathcal{H} \times \{\pm 1\}^M$ and every realization of the noise $\boldsymbol{\alpha}$.*

*Proof.* We will prove this lemma by induction on $T$. For the base case, $T = 0$, and so it suffices to show that $\boldsymbol{\alpha} \cdot (\boldsymbol{\Gamma}_{(h_1, \boldsymbol{\theta}_1)} - \boldsymbol{\Gamma}_{(h, \boldsymbol{\theta})}) + \epsilon \geq 0$ for all $(h, \boldsymbol{\theta})$. Here, the statement follows directly from the $\epsilon$-approximate optimality of oracle; that is, we select $(h_1, \boldsymbol{\theta}_1)$ such that

$$\boldsymbol{\alpha} \cdot \boldsymbol{\Gamma}_{(h_1, \boldsymbol{\theta}_1)} \geq \max_{h \in \mathcal{H}, \boldsymbol{\theta} \in \{\pm 1\}^M} \left\{ \boldsymbol{\alpha} \cdot \boldsymbol{\Gamma}_{(h, \boldsymbol{\theta})} \right\} - \epsilon.$$

Next, we prove the inductive step. Assume that the lemma holds for some $T$. Now, for all $(h, \boldsymbol{\theta}) \in \mathcal{H} \times \{\pm 1\}^M$, we have

$$\sum_{t=1}^{T+1} \langle \boldsymbol{\theta}_{t+1}, h_{t+1}(\mathbf{x}_t) \cdot \boldsymbol{f}_t \rangle + \boldsymbol{\alpha} \cdot \boldsymbol{\Gamma}_{(h_1, \boldsymbol{\theta}_1)} = \sum_{t=1}^T \langle \boldsymbol{\theta}_{t+1}, h_{t+1}(\mathbf{x}_t) \cdot \boldsymbol{f}_t \rangle + \boldsymbol{\alpha} \cdot \boldsymbol{\Gamma}_{(h_1, \boldsymbol{\theta}_1)} + \langle \boldsymbol{\theta}_{T+2}, h_{T+2}(\mathbf{x}_{T+1}) \cdot \boldsymbol{f}_{T+1} \rangle$$

$$\geq \sum_{t=1}^T \langle \boldsymbol{\theta}_{T+2}, h_{T+2}(\mathbf{x}_t) \cdot \boldsymbol{f}_t \rangle + \boldsymbol{\alpha} \cdot \boldsymbol{\Gamma}_{(h_{T+2}, \boldsymbol{\theta}_{T+2})} - \epsilon \cdot (T + 1)$$

$$\quad + \langle \boldsymbol{\theta}_{T+2}, h_{T+2}(\mathbf{x}_{T+1}) \cdot \boldsymbol{f}_{T+1} \rangle$$

$$= \sum_{t=1}^{T+1} \langle \boldsymbol{\theta}_{T+2}, h_{T+2}(\mathbf{x}_t) \cdot \boldsymbol{f}_t \rangle + \boldsymbol{\alpha} \cdot \boldsymbol{\Gamma}_{(h_{T+2}, \boldsymbol{\theta}_{T+2})} - \epsilon \cdot (T + 1)$$

$$\geq \sum_{t=1}^{T+1} \langle \boldsymbol{\theta}, h(\mathbf{x}_t) \cdot \boldsymbol{f}_t \rangle + \boldsymbol{\alpha} \cdot \boldsymbol{\Gamma}_{(h, \boldsymbol{\theta})} - \epsilon \cdot (T + 2)$$

where the first inequality follows by applying the induction hypothesis and considering the pair $(h_{T+2}, \boldsymbol{\theta}_{T+2})$, and the final inequality follows by the $\epsilon$-approximate optimality of the oracle at time $T + 2$. This completes the proof of the lemma. $\square$

Next, we characterize the term $T_2$ in (19). The term $T_2$ is essentially a *stability error term*, which measures the cumulative effect of the difference in the decisions $(h_{t+1}, \boldsymbol{\theta}_{t+1})$ and $(h_t, \boldsymbol{\theta}_t)$ on reward. Intuitively, a stable algorithm will result in similar or slowly varying decisions over time and therefore a small stability error term. The following lemma characterizes the stability error term under the condition that the perturbation matrix $\boldsymbol{\Gamma}$ is $\delta$-admissible.

**Lemma D.2.** *(Stability Lemma) Suppose we run Algorithm 3 with a $\delta$-admissible matrix $\boldsymbol{\Gamma} \in [0, 1]^{(|\mathcal{H}| \times 2^M) \times N}$ and random vector $\boldsymbol{\alpha} = (\alpha_1, \cdots, \alpha_N)$ such that each $\alpha_i$ is drawn independently from $\text{Unif}[0, \sqrt{T}]$. Then, we have*

$$T_2 \;\leq\; 4TN \cdot (1 + \delta^{-1}) \cdot \left( \frac{B + \epsilon}{\sqrt{T}} \right).$$

The proof is identical to the proof of the stability lemma (Lemma 2.4) presented in (Dudík et al., 2020), and is omitted here. Plugging our upper bounds on $T_1$ and $T_2$ into (19) gives us

$$
\begin{aligned}
R_T(h) &\leq \mathbb{E}\left[ \boldsymbol{\alpha} \cdot \left( \boldsymbol{\Gamma}_{(h_1, \boldsymbol{\theta}_1)} - \boldsymbol{\Gamma}_{(h, \boldsymbol{\theta}_h^*)} \right) \right] + \epsilon \cdot (T + 1) + 4TN \cdot (1 + \delta^{-1}) \cdot \left( \frac{B + \epsilon}{\sqrt{T}} \right) \\
&\leq \mathbb{E}\left[ \boldsymbol{\alpha} \cdot \boldsymbol{\Gamma}_{(h_1, \boldsymbol{\theta}_1)} \right] + \epsilon \cdot (T + 1) + 4TN \cdot (1 + \delta^{-1}) \cdot \left( \frac{B + \epsilon}{\sqrt{T}} \right) \\
&\leq N\sqrt{T} + \epsilon \cdot (T + 1) + \frac{16BN\sqrt{T}}{\delta} = O\left( \frac{BN\sqrt{T}}{\delta} \right)
\end{aligned}
$$

Above, the first inequality uses the fact that $\boldsymbol{\alpha} \succeq \mathbf{0}$ and $\boldsymbol{\Gamma} \in [0, 1]^{(|\mathcal{H}| \times 2^M) \times N}$; the second inequality uses the fact that $\boldsymbol{\Gamma} \in [0, 1]^{(|\mathcal{H}| \times 2^M) \times N}$ and each $\alpha_i$ is drawn independently from $\text{Unif}[0, \sqrt{T}]$, and the final equality uses the assumption that $\epsilon = 1/\sqrt{T}$.

Finally, we argue that if $\boldsymbol{\Gamma}$ satisfies implementability with complexity $D$, then Algorithm 3 with $\epsilon = 1/\sqrt{T}$ can be implemented in time $poly(N, D, T)$. This directly follows by noting that for any $\boldsymbol{\alpha}$, we have a set of coefficients $\{\kappa_j\}_{j=1}^D$ and vectors $\{\widehat{\boldsymbol{f}}_j\}_{j=1}^D$ (which may depend on $\boldsymbol{\alpha}$) such that

$$
\sum_{s=1}^{t-1} \langle \boldsymbol{\theta}, h(\mathbf{x}_s) \cdot \boldsymbol{f}_s \rangle + \boldsymbol{\alpha} \cdot \boldsymbol{\Gamma}_{(h, \boldsymbol{\theta})} = \sum_{s=1}^{t-1} \langle \boldsymbol{\theta}, h(\mathbf{x}_s) \cdot \boldsymbol{f}_s \rangle + \sum_{j=1}^{D} \kappa_j \cdot \left\langle \boldsymbol{\theta}, h(\mathbf{x}_j) \cdot \widehat{\boldsymbol{f}}_j \right\rangle.
$$

This is exactly in the form of our offline oracle (Definition 4.1). Thus, the runtime of Algorithm 3, treating as black boxes the step of sampling from the uniform distribution and the oracle (3) is $O(T^2 + TND)$. This completes the proof of Theorem 4.4.

### D.2. Proof of Lemma 4.5

*Proof.* We consider $\boldsymbol{\Gamma} \in [-B, B]^{(|\mathcal{H}| \times 2^M) \times (DM)}$, so that the rows of $\boldsymbol{\Gamma}$ can be indexed by $(h, \boldsymbol{\theta}) \in \mathcal{H} \times \{\pm 1\}^M$ and the columns of $\boldsymbol{\Gamma}$ can be indexed by $(j, i) \in [D] \times [M]$. Then, we will specify each entry as $\boldsymbol{\Gamma}_{(h, \boldsymbol{\theta}), (i, j)} = h(\mathbf{x}^i) \cdot \theta_j$. Note that $|\boldsymbol{\Gamma}_{(h, \boldsymbol{\theta}), (i, j)}| \leq B$ since $h \in \mathcal{H} \subseteq \mathcal{H}_B$. We first prove implementability and then 1-admissibility.

**Proof of implementability.** Consider any realization of the noise $\boldsymbol{\alpha}$. As with the columns of $\boldsymbol{\Gamma}$, we index $\boldsymbol{\alpha}$ by the tuple $(j, i) \in [D] \times [M]$. We denote as $\boldsymbol{\alpha}_{(j, \cdot)} \in \mathbb{R}^M$ the sub-vector $\left( \alpha_{(j, 1)}, \cdots, \alpha_{(j, M)} \right)$. Then, we have

$$
\begin{aligned}
\langle \boldsymbol{\alpha}, \boldsymbol{\Gamma}_{(h, \boldsymbol{\theta})} \rangle &= \sum_{j=1}^{D} \sum_{i=1}^{M} \alpha_{(j, i)} \cdot h(\mathbf{x}^j) \cdot \theta_i \\
&= \sum_{j=1}^{D} h(\mathbf{x}^j) \langle \boldsymbol{\alpha}_{(j, \cdot)}, \boldsymbol{\theta} \rangle,
\end{aligned}
$$

which clearly satisfies the implementability condition (5) with complexity $D$, $\kappa_j = 1$ for all $j \in [D]$, and $\widetilde{\boldsymbol{f}}_j = \boldsymbol{\alpha}_{(j, \cdot)}$.

**Proof of admissibility.** Consider any two rows indexed by $(h, \boldsymbol{\theta})$, $(h', \boldsymbol{\theta}')$. We need to show that:

1. $\mathbf{\Gamma}_{(h,\boldsymbol{\theta})} \neq \mathbf{\Gamma}_{(h',\boldsymbol{\theta}')}$.

2. If $\Gamma_{(h,\boldsymbol{\theta}),(j,i)} \neq \Gamma_{(h',\boldsymbol{\theta}'),(j,i)}$, then $|\Gamma_{(h,\boldsymbol{\theta}),(j,i)} - \Gamma_{(h',\boldsymbol{\theta}'),(j,i)}| \geq 1$.

Showing the first statement is easy: either $h \neq h'$, in which case there exists at least one index $j \in [D]$ such that $h(\mathbf{x}^j) \neq h'(\mathbf{x}^j)$. In the other case where $h = h'$, consider any index $j$ such that $h(\mathbf{x}^j) \neq 0$ (at least one such index must exist, otherwise we can simply remove $h$ from the hypothesis class without loss of generality). Further, since in this case we must have $\boldsymbol{\theta} \neq \boldsymbol{\theta}'$, there exists at least one index $i$ for which $\theta_i \neq \theta_i'$. Therefore, we have $\mathbf{\Gamma}_{(h,\boldsymbol{\theta}),(j,i)} = h(\mathbf{x}^j)\theta_i \neq h(\mathbf{x}^j)\theta_i'$.

To show the second statement, we use the structure of the Boolean hypercube to say that if $\theta_i \neq \theta_i'$, then $|\theta_i - \theta_i'| = 2$. Similarly, if $\theta_i \neq 0$, then $|\theta_i| = 1$. Then, we again consider two cases when $\mathbf{\Gamma}_{(h,\boldsymbol{\theta}),(j,i)} \neq \mathbf{\Gamma}_{(h',\boldsymbol{\theta}'),(j,i)}$. The first is if $h(\mathbf{x}^j) \neq h'(\mathbf{x}^j)$. In this case, suppose without loss of generality that $h(\mathbf{x}^j) = 1$ while $h'(\mathbf{x}^j) = 0$. Then, we have

$$|\mathbf{\Gamma}_{(h,\boldsymbol{\theta}),(j,i)} - \mathbf{\Gamma}_{(h',\boldsymbol{\theta}'),(j,i)}| = |\theta_i| = 1.$$

On the other hand, suppose that $h(\mathbf{x}^j) = h'(\mathbf{x}^j)$. If $h(\mathbf{x}^j) = 0$, we have $\Gamma_{(h,\boldsymbol{\theta}),(j,i)} = \Gamma_{(h',\boldsymbol{\theta}'),(j,i)}$. If $h(\mathbf{x}^j) = 1$, we have

$$|\mathbf{\Gamma}_{(h,\boldsymbol{\theta}),(j,i)} - \mathbf{\Gamma}_{(h',\boldsymbol{\theta}'),(j,i)}| = |\theta_i - \theta_i'| \in \{0, 2\}$$

This proves the second statement and therefore we have proved 1-admissibility, which completes the proof of the lemma. $\quad\square$

