# OpenReview forum: "Improved and Oracle-Efficient Online $\ell_1$-Multicalibration"
_ICML.cc/2025/Conference — ICML 2025 poster_

### Official Review · Reviewer_4b8V · 2025-03-05

**Overall Recommendation:** 3

**Summary:**

This paper tackles the challenge of online multicalibration. The key contribution of this paper is theoretical: the paper proposes a method that achieves improved rate of O(T^-1/3) and oracle efficient rate of O(T^-1/4). The key insight is that one can reduce the L1 multicalibration problem into an online linear-product optimization problem (OLPO).

**Claims And Evidence:**

Yes, claims are usually backed up with references or motivation.

**Essential References Not Discussed:**

NA

**Experimental Designs Or Analyses:**

No experiments.

**Methods And Evaluation Criteria:**

The paper does not provide any evaluation of the proposed approach.

**Other Comments Or Suggestions:**

Overall, I don't feel like the community at ICML could benefit from these theoretical results and probably another venue could be a better fit. However, I am open to change my mind, should the other reviewers be positive on the paper.

**Other Strengths And Weaknesses:**

# Main Strengths:

1. The paper tackles the multicalibration problem, which is a nice property that models should've to output precise probabilities. By analyzing theoretically the proposed solution, the paper improves the convergence rate;

2. Several results are provided, showcasing the importance of online multicalibration;

# Main Weaknesses:

1. The paper is extremely heavy on theory. I wonder whether ICML is the proper venue to present such findings or another conference (e.g., theoretical computer science) could be a better fit;

2. The structure of the paper is not ideal: first results are provided (Sec 1), and after they are proved (Sec 2). Also, I do not understand how 1.1 and the next subsections could fit into the introduction. I'd rather have a Sec 2 starting from 1.1;

3. Although the paper presents a pseudo-code for the proposed algorithm, seeing results on toy data would improve the quality of the paper;

4. The motivation underlying the paper is hidden, and it seems to simply be "this existing paper achieves the following guaranteed error, can we do better?". The point of why this is relevant should be highlighted.

**Questions For Authors:**

NA

**Relation To Broader Scientific Literature:**

The paper improves upon existing literature by improving the oracle efficient rate from T^-1/8 to T^-1/4, and the online multicalibration rate from T^-1/4 to T^-1/3. Also, the paper mentions connections with omnipredictions.

**Theoretical Claims:**

The paper is extremely heavy on theory and I did not check the correctness of proofs.

---

> ### Author Rebuttal · Authors · 2025-03-31
>
> We appreciate the reviewer’s concern regarding the theoretical nature of our work. While our contributions are indeed theoretical, we believe ICML is an appropriate venue for the following reasons. (1) Relevance to Core ML Problems: Multicalibration — and online multicalibration in particular — is a key problem at the intersection of machine learning, fairness, and sequential decision-making. Recent ICML papers (e.g., on online learning, calibration, multigroup fairness, omniprediction) have explored similar themes, indicating strong interest from the ICML community.
> (2) Methodological Advancement: Our work introduces a new online learning problem (OLPO) — and gives a reduction similar to one that was obtained for online calibration and OLO.
> (3) Precedent: Similar works were published at ICML; see, e.g., Noarov and Roth (2023), Globus-Harris et al. (2023). In fact, the original multicalibration paper, Hébert-Johnson et al. (2018), also appeared in ICML.
> Works on online learning, optimization and oracle-efficient online learning also regularly appear at ICML.
>
> $Regarding \ the \ motivation$: While our work does improve upon existing error rates, our motivation is broader: online multicalibration is a key tool for ensuring fairness and reliability in sequential prediction settings, which arise in many real-world applications. Our contributions aim to make this framework both statistically and computationally efficient. We will revise the introduction to better highlight this broader motivation and its relevance to the ML community.

---

### Official Review · Reviewer_uPiD · 2025-03-14

**Overall Recommendation:** 4

**Summary:**

The paper focuses on the online multicalibration task, for which it (1) presents a O(T^{2/3})-ECE error algorithm, thus matching the best known constructive efficient bounds for vanilla calibration; and (2) presents an oracle-efficient algorithm that obtains O(T^{3/4}) multicalibration ECE error given access to a certain offline oracle.

Both results above are achieved by developing a new online prediction framework that the authors called online linear product optimization (OLPO), which is distinct from online convex optimization. They also develop a linearized variant of OLPO, which helps obtain the first contribution.  They obtain the second contribution via implementing (non-linearized) OLPO via the oracle-efficient FTPL technique of Dudik et al (2020), which works under two certain conditions. They then show that for the natural "transductive" and "small-separator" classes of group families, these conditions always hold.

#######
Update after rebuttal:
I have read the authors' response. It addresses my concerns by promising to include additional discussion of the reference, and I therefore keep my original score.

**Claims And Evidence:**

Yes, the claims made in the submission are supported by convincing evidence in the form of proofs.

**Essential References Not Discussed:**

The following paper [NRRX'23] (https://opt-ml.org/papers/2023/paper96.pdf) contains what appears to be a highly related framework for online unbiased vector-valued prediction, where in their case unbiasedness is stated with respect to general events that may depend on the context and on the predictions themselves. In particular, their method already attains O(T^{2/3}) L1 multicalibration as an immediate corollary. Namely, discretizing into m uniform buckets and defining conditioning events for each pair (group, bucket), their framework gives, for each group after T rounds, L1 calibration on every group bounded as: T/m (accumulated discretization error over the T rounds) + O(\sum_{buckets i} \sqrt{number of rounds prediction fell into bucket i}). In the worst case when each bucket appears in T/m rounds this leads to the bound O(T/m + \sqrt{Tm}) = O(T^{2/3}) for the standard choice m = T^{1/3}.

Beyond this result, it appears that the OLPO is quite related to this unbiased framework, with both reducing to applying the combination of experts + OCO algorithms, so it appears important for context to discuss this connection in some detail.

**Experimental Designs Or Analyses:**

N/A --- this paper presents a theoretical contribution only.

**Methods And Evaluation Criteria:**

N/A --- this paper presents a theoretical contribution only.

**Other Comments Or Suggestions:**

A presentational suggestion: I believe the derivation in the last appendix, which confirms the two conditions in the case of transductive and small separator set groups, is actually quite important for the reader to ingest, so I think it is wise to move it to the main part of the paper --- indeed, it nicely clarifies the applicability of the novel oracle-efficient construction.

**Other Strengths And Weaknesses:**

Overall, I believe that the main strength of this paper lies in its carefully built framework that allows to exploit the oracle-efficient FTPL technique in this setting. While oracle efficient FTPL is clearly a possible candidate subroutine to guarantee oracle efficiency in this and many other online settings, it remained unclear in the online calibration literature until now how to connect this setting so that this subroutine can be exploited. This paper does that, and also makes a first step to identify: under what conditions can group families induce this efficient oracle algorithm? Similarly, also instructive and fruitful are the careful techniques exploiting the calibration halfspace oracle of Abernethy et al (2011) that are involved in the first part of the paper. Another strength is that the paper is well-written.

**Questions For Authors:**

The main question that I have is a substantive comparison to the reference above --- beyond the O(T^{2/3}) L1 multicalibration implication, OLPO appears to share some essential algorithmic features with the proposed framework here, seeing as both frameworks rely on appropriately linearizing the task into a combination of an experts algorithm to aggregate over the "groups", and on an online convex optimization method to form actual predictions.

**Relation To Broader Scientific Literature:**

There exists a large prior literature on online vanilla calibration, and a smaller but substantial literature on online multicalibration and related methods. Relative to this latter literature, this paper obtains new algorithms for online multicalibration with rates matching the (near-) best known rates for vanilla calibration, as well as the first oracle efficient algorithm.

For the latter, oracle efficiency, part, the closest-related prior work is (Garg et al, 2024), which obtained oracle efficient online omniprediction (a closely related but distinct task), but the sense in which Garg et al's was an oracle-efficient algorithm required access to an online (regression) oracle, whereas the present paper shows in its setting that an offline oracle can suffice, and do so via novel arguments in their developed framework, connecting it to the oracle efficiency afforded by an adaptive FTPL algorithm of Dudik et al.

For the former contribution, that is, the O(T^{2/3}) L1 multicalibration rate, I believe this rate is already subsumed by an existing online unbiased prediction framework (https://opt-ml.org/papers/2023/paper96.pdf); see below. The OLPO algorithm and framework used by this paper to obtain this result is similar but appears distinct from the former framework (the two frameworks might possibly be duals of each other in some sense).

**Theoretical Claims:**

Yes, I checked the bulk of proofs and statements in the main part and in the supplementary, and believe them to be correct --- possibly modulo some small unchecked technicalities.

---

> ### Author Rebuttal · Authors · 2025-03-31
>
> We thank the reviewer for pointing us towards this relevant reference.
> After reviewing their results, we agree that their framework can be used to derive bounds for online $\ell_1$-multicalibration, and we outline a high-level approach for binary-valued hypothesis classes below.
> Fix a $h \in \mathcal H$ and denote the collection of events
> $\mathcal E := \{E_{h,i}: h \in \mathcal H \text{ and } i \in \{0,\ldots,m\} \}$
> where $E_{h,i}(x,p) = \mathbb{I}[h(x_t) = 1, p_t = i/m]$. Further, denote $T_{h,i} := \sum_{t=1}^T \mathbb{I}[h(x_t) = 1, p_t = i/m]$ and note that this is equal to $\sum_{t=1}^T E_{h,i}(x_t,p_t)$. Also note that $\sum_{i = 0}^m T_{h,i} \leq T$. Then, applying Theorem~3.4 of the reference together with the halfspace oracle, one would obtain
> $$
> K_T(\pi,h) \lesssim \sum_{i = 0}^m \sqrt{T_{h,i} \cdot \log (2|\mathcal H|mT)} + \frac{T}{m} \leq \sqrt{mT \cdot \log (2|\mathcal H|mT)} + \frac{T}{m},
> $$
> and then optimizing over the discretization level $m$ gives the result.
>
> It is worth noting that the algorithmic framework in the paper [NRRX'23] is quite different, despite both works using an expert routine. In particular, the work [NRRX'23] requires a small-loss regret bound to get the result, while we do not.
> Additionally, we believe our algorithm is much simpler (e.g. not requiring the solution of any min-max optimization problem) and the reduction to OLPO is of independent interest as it facilitates the oracle-efficient results in a more natural and modular way.
> We will certainly include a discussion of this connection in the related work section to provide better context and contrast our contributions with theirs.
>
> Finally, thank you for the presentational feedback. We agree and we would be happy to move this derivation (at least the statements of the conditions, if not the full proof) to the main body of the paper.

---

### Official Review · Reviewer_avZG · 2025-03-14

**Overall Recommendation:** 4

**Summary:**

The paper studies the problem of online multicalibration for L1 norm. The paper proposes a method with theoretical guarantees. The key contribution is based on the reduction of online L1-multicalibration to an online learning problem.

### update after rebuttal
I am maintaining the current score following the rebuttal.

**Claims And Evidence:**

These claims are well-supported through theoretical analyses and mathematical proofs.

**Essential References Not Discussed:**

N/A

**Experimental Designs Or Analyses:**

N/A

**Methods And Evaluation Criteria:**

N/A

**Other Comments Or Suggestions:**

N/A

**Other Strengths And Weaknesses:**

1. The paper is very well-written and the maths appears to be rigorous.
2. The paper proposed an approach that only takes a polynomial number of call to an optimization oracle, greatly improve the no-regret algorithm for OLPO.

**Questions For Authors:**

N/A

**Relation To Broader Scientific Literature:**

The results improve the prior work on online multicalibration (Gupta 2024, Garg 2024).   In particular, the error rate when H is finite is improved from $O(T^{-1/4})$ in Garg 2024 to  $O(T^{-1/3})$ in this paper. Morever, the oracle-efficient bound is also improved from  $O(T^{-1/8})$ in Garg 2024 to  $O(T^{-1/4})$.

**Theoretical Claims:**

The theoretical contributions are sound. The idea on extending the algorithms to a more general setting make sense. The proposed OLPO is novel.

---

> ### Author Rebuttal · Authors · 2025-03-31
>
> We thank the reviewer for their thoughtful and positive evaluation of our paper.

---

### Official Review · Reviewer_F47Q · 2025-03-20

**Overall Recommendation:** 3

**Summary:**

The paper studies an online prediction setting, where a learner wished to minimize $\ell_1$ multicalibration error with respect to a class of real-valued predictors $\mathcal{H}$ that act as group selection functions. The authors propose an algorithm that obtains an error rate of $O(T^{-1/3})$, through reducing the problem to an online problem they term online linear product minimization, which they solve by an linearizing the reward function by enumerating over all $h\in\mathcal{H}$, and running an algorithm that employs an algorithm for the expert setting over $\vert\mathcal{H}\vert$ online linear optimization algorithms ("meta-experts"), one for each $h\in\mathcal{H}$. They then show how to extend their result to non-finite classes $\mathcal{H}$ using the technique of covering numbers and finding a finite cover for $\mathcal{H}$. Finally, the authors consider obtaining online $\ell_1$ multicalibration with oracle-efficient algorithms (which do not require enumerating over all of $\mathcal{H}$, but instead assume access to an offline optimization oracle), for which they show an algorithm based on a reduction to the generalized FTPL framework (Dudik et al. 2020), and obtaining an error rate of $O(T^{-1/4})$.

**Claims And Evidence:**

The paper is purely theoretical. I have only glanced at the appendix with the proofs, but the claims appear to be substantiated.

**Essential References Not Discussed:**

Most of the relevant related work appears in in manuscript. An exception is "High-Dimensional Prediction for Sequential Decision Making" by Noarov et al. 2023 that I believe is highly relevant and is not discussed. I believe that using their approach of obtaining subsequence regret guarantees can be utilized when each subsequence is defined by $h\in\mathcal{H}$, and optimizing over the number of buckets in the prediction should obtain similar bounds of $O(T^{-1/3})$.

**Experimental Designs Or Analyses:**

N/A.

**Methods And Evaluation Criteria:**

N/A.

**Other Comments Or Suggestions:**

-

**Other Strengths And Weaknesses:**

Strengths:
1. The paper is very well-written and presentation is clear and concise. I enjoyed reading it.
2. The problem is clearly motivated, and is rather central in the domains of fairness/uncertainty estimation.
3. The result on obtaining $O(T^{-1/4})$ $\ell_1$ multicalibration with an oracle-efficient algorithm and an offline oracle I believe is novel and very interesting, as well as the bounds for non-finite classes in the first part of the paper.

Weaknesses:
1. Novelty of the $O(T^{-1/3})$ bound using an inefficient algorithm in the finite case, please see questions section.

**Questions For Authors:**

Can the approach in Noarov et al. 2023 for subsequence regret be utilized to derive similar bounds of $O(T^{-1/3})$ for $\ell_1$ multicalibration? Can you elaborate on the differences?

**Relation To Broader Scientific Literature:**

The paper is studying online $\ell_1$ multicalibration and provides: (a) (computationally inefficient) algorithm that improves error rates over prior work (Gupta et al. 2022, Lee et al. 2022) (though Noarov et al. may implicitly still obtain the same bounds, see question to authors) and extensions to non-finite classes, and (b) oracle-efficient algorithm obtaining a faster error rate than known in prior work (Garg et al. 2024), and under milder assumptions (offline oracle instead of online regression oracle).

**Theoretical Claims:**

I did not.

---

> ### Author Rebuttal · Authors · 2025-03-31
>
> We thank the reviewer for their positive evaluation of our paper, and for pointing us towards the paper [NRRX'23]. Please refer to the response to Reviewer~uPiD.

---

### Official Review · Reviewer_FERs · 2025-03-23

**Overall Recommendation:** 4

**Summary:**

This paper studies the online l1-multicalibration problem. Multicalibration is a natural extension of calibration with group identities. It is a natural group fairness definition and implies some learning concept called omniprediction.

The authors improved based on a previous work that provides O(T^{1/4}) upperbound for l2-multicalibration. This paper directly solves l1-multicalibration with O(T^{1/3}) and oracle efficient O(T^{1/4}) upperbound using halfspace oracle, which they provided an implementation in the paper. This paper improves the previous work in many perspectives.

**Claims And Evidence:**

Yes.

**Essential References Not Discussed:**

N/A

**Experimental Designs Or Analyses:**

N/A

**Methods And Evaluation Criteria:**

Yes

**Other Comments Or Suggestions:**

There is one typo that needs to be fixed at the end of Appendix C but does not hurt understanding.

**Other Strengths And Weaknesses:**

This paper provides a reduction from online multicalibration to online linear-product optimization, which is similar to the connection between calibration and online linear optimization. The result is an improvement, and the reduction provides some new understanding of the online multicalibration problem.

**Questions For Authors:**

I am wondering if the authors can confirm that their results also provide the same upperbound for online omniprediction for free since l1-multicalibration error is an upper bound for omniprediction error in the offline setting.

Also, does the lower bound in online expected calibration error also a lower bound for online multicalibration? I think it would be nice if the authors could discuss a bit in related work.

**Relation To Broader Scientific Literature:**

This paper provides some better results on a problem studied by some previous work.

**Theoretical Claims:**

I read the arguments at a high level, and they make sense to me.

---

> ### Author Rebuttal · Authors · 2025-03-31
>
> We thank the reviewer for their thorough review of our paper (and for discovering a typo in Appendix~C). We also appreciate the reminder to acknowledge the known lower bound for online calibration, which indeed extends to online $\ell_1$- multicalibration.
>
> To address the first comment, yes, our results do imply improved bounds for online omniprediction since our improved bounds for online $\ell_1$-multicalibration can be transformed into improved bounds for online omniprediction when the loss functions are convex and Lipschitz; see,  e.g., Garg et al. (2024).
> We opted against including results for online omniprediction because it was not the primary focus of our work, and we wanted to avoid distracting from the core contributions on online $\ell_1$-multicalibration and the $\mathtt{OLPO}$ framework.
> Additionally, due to space limitations, providing a satisfactory treatment of omniprediction results would have been challenging.
> That said, we acknowledge that this connection is standard and well-known in the community, and we will add a short discussion in the related work in the final version of the paper.

---

### Decision · Program_Chairs · 2025-05-01

**Decision:**

Accept (poster)

**Comment:**

The paper considers the online multicalibration problem, and develops T^(2/3) and T^(3/4) regret algorithms for the oracle-inefficient and oracle-efficient case. The reviewers pointed out that the T^(2/3) regret for the oracle-inefficient case can be achieved by a recent work of Noarov et al. 2023. However, this does not detract that much from the quality of the paper, since the proposed algorithm is very different. In addition, the oracle-efficient guarantee is significant and novel. The results should be of interest to the ICML audience, though I recommend that the authors rewrite things a bit to make it more readable for a broader audience.